# Modeling tissue-relevant *Caenorhabditis elegans* metabolism at network, pathway, reaction, and metabolite levels

Lutfu Safak Yilmaz[1,*,†] (iD), Xuhang Li[1,†], Shivani Nanda[1], Bennett Fox[2], Frank Schroeder[2] &
Albertha JM Walhout[1,**] (iD)

## Abstract

**Metabolism is a highly compartmentalized process that provides building blocks for biomass generation during development, homeostasis, and wound healing, and energy to support cellular and organismal processes. In metazoans, different cells and tissues specialize in different aspects of metabolism. However, studying the compartmentalization of metabolism in different cell types in a whole animal and for a particular stage of life is difficult. Here, we present MEtabolic models Reconciled with Gene Expression (MERGE), a computational pipeline that we used to predict tissue-relevant metabolic function at the network, pathway, reaction, and metabolite levels based on single-cell RNA-sequencing (scRNA-seq) data from the nematode *Caenorhabditis elegans*. Our analysis recapitulated known tissue functions in *C. elegans*, captured metabolic properties that are shared with similar tissues in human, and provided predictions for novel metabolic functions. MERGE is versatile and applicable to other systems. We envision this work as a starting point for the development of metabolic network models for individual cells as scRNA-seq continues to provide higher-resolution gene expression data.**

**Keywords** *Caenorhabditis elegans*; data integration; metabolic network; single-cell RNA-seq; tissue metabolism

**Subject Categories** Computational Biology; Metabolism; Methods & Resources

**Mol. Syst. Biol. (2020) 16: e9649**

## Introduction

Metabolism can be studied at a systems level using genome-scale metabolic networks that describe the total collection of metabolic reactions required for the generation of biomass and energy, and the general maintenance of homeostasis (O'Brien *et al*, 2015; Angione, 2019). A systems-level understanding of metabolism in complex multicellular organisms requires the reconstruction of metabolic networks at the level of different tissues and, ultimately, individual cells. To understand the function of metabolic networks at the level of individual cells or tissues, it is important to know which parts of the whole network are active in each cell or tissue and which parts are inactive. Single-cell or tissue-level protein expression and enzyme activity data are often not available. However, we and others have shown that mRNA levels provide a powerful proxy to construct context-relevant metabolic network models (Machado & Herrgard, 2014; Robaina Estevez & Nikoloski, 2014; Yilmaz & Walhout, 2016).

The nematode *Caenorhabditis elegans* is a hermaphrodite that develops from embryos through four larval stages to adults via a deterministic lineage. Adult *C. elegans* are comprised of 959 somatic nuclei that form the major tissues, such as muscle, intestine, and hypodermis (skin). *Caenorhabditis elegans* is a bacterivore that can be fed individual bacterial strains. *Caenorhabditis elegans* tissues and metabolism share many functions with mammals. Therefore, it provides a relatively simple model for understanding animal metabolism at a systems level. We have previously reconstructed a *C. elegans* genome-scale metabolic network model (Yilmaz & Walhout, 2016), which we validated using flux balance analysis (FBA) (Raman & Chandra, 2009).

Metabolic network models and gene expression data can be integrated at the network level qualitatively, semi-quantitatively, or quantitatively. Qualitative methods typically define context-specific networks by excluding reactions that are not associated with highly expressed genes (Jerby *et al*, 2010; Agren *et al*, 2012; Wang *et al*, 2012; Vlassis *et al*, 2014). Semi-quantitative approaches predict the metabolic state in the form of a flux distribution that avoids flux in reactions associated with lowly expressed genes and may divert flux to reactions associated with highly expressed genes (Becker & Palsson, 2008; Zur *et al*, 2010). Quantitative integration methods that can model tissue metabolism

1 Program in Systems Biology, Program in Molecular Medicine, University of Massachusetts Medical School, Worcester, MA, USA
2 Boyce Thompson Institute, Department of Chemistry and Chemical Biology, Cornell University, Ithaca, NY, USA
*Corresponding author. Tel: +1 508 856 6711; E-mail: lutfu.yilmaz@umassmed.edu
**Corresponding author. Tel: +1 508 856 3601; E-mail: marian.walhout@umassmed.edu
†These authors contributed equally to this work as first authors

have also been developed (Brandes *et al*, 2012; Navid & Almaas, 2012; Pandey *et al*, 2019). Such methods fit flux distributions to expression data in a continuous fashion. However, these methods typically depend on a selected objective function such as biomass production, and it is not feasible to capture all metabolic functions with a single objective function.

Here, we developed a new computational pipeline we name MERGE (MEtabolic models Reconciled with Gene Expression), a combined approach (Fig 1A) that starts with a semi-quantitative, network-level integration, evaluates the variability of the resulting flux distribution to obtain tissue-specific metabolic networks, and finally uses these networks to quantitatively integrate local gene expression data at the pathway level to provide relative flux predictions at the reaction and metabolite levels. We used MERGE to study tissue metabolism in the nematode *C. elegans* based on scRNA-sequencing data obtained at the second larval stage (L2) (Cao *et al*, 2017). We derived functional metabolic network models of seven major tissues for which transcriptomes were generated in the reference study (Cao *et al*, 2017) by aggregating scRNA-seq data from thousands of individual cells that originate from the same tissue in a population of animals. Our results recapitulate known tissue functions, reveal metabolic properties that are shared with similar tissues in human, and predict numerous novel metabolic functions. MERGE provides a versatile tool for the integration of high-quality gene expression data with genome-scale metabolic network models that provides an important step toward the quantitative modeling of metabolism at the level of individual cells.

## Results

### Updates in *C. elegans* metabolic network reconstruction

Prior to deriving tissue-relevant metabolic networks, we updated our previous model of *C. elegans* metabolism (Yilmaz & Walhout, 2016) named iCEL1273 (Fig 1B, Tables EV1–EV3). The updated model, which we refer to as iCEL1314, retains a similar structure to the original model and the additions did not affect major functions of the network (Yilmaz & Walhout, 2016). The updates include the reconstruction of an ascaroside biosynthesis pathway that produces the most abundant ascarosides (von Reuss *et al*, 2012; Zhang *et al*, 2015; Artyukhin *et al*, 2018) (Fig EV1, Appendix Supplementary Methods), the incorporation of new transport reactions based on the

recently updated human metabolic network model Recon 3D (Brunk *et al*, 2018), two genes and four reactions from ElegCyc (Gebauer *et al*, 2016), and different types of manual curations varying from the elimination of pseudo- and dead genes based on WormBase (Harris *et al*, 2013) to the modification of the fatty acid composition of sphingolipids (Witting *et al*, 2018). iCEL1314 contains 1,314 genes, 2,230 reactions, and 907 unique metabolites (Fig 1C, Tables EV2 and EV3).

### Processing of the gene expression dataset

For the generation of tissue-relevant metabolic network models, we selected a high-quality scRNA-seq dataset of the L2 stage that was used to derive aggregated transcriptomes of seven major tissues (Cao *et al*, 2017) (Fig 1D). In the first step of MERGE, we developed a semi-quantitative approach where genes are divided into four categories for each tissue: highly, moderately, lowly, and rarely expressed (Fig 1C, Table EV4). In a tissue, flux is encouraged in reactions associated with genes that are highly expressed, while reactions dependent on lowly or rarely expressed genes are discouraged from carrying flux. Reactions associated with moderately expressed genes are left free, so they carry flux only if the flux in the rest of the network requires them to (Fig 1C). To place genes into different categories, we used a statistical analysis of gene expression, rather than arbitrary cutoffs (Fig EV2 and Appendix Fig S1, Materials and Methods). In addition, we developed a heuristic algorithm to recategorize moderately expressed genes as highly or lowly expressed if they are enriched or depleted in some tissues relative to others (Fig EV3). The distribution of the four gene expression categories was similar for each tissue (Fig 1E). However, when all tissues were combined, we found that the majority of metabolic genes are highly expressed in at least one tissue (Fig 1E). These results indicate that most of the metabolic network is active somewhere in L2 animals and that a large portion of the network is enriched or depleted in different tissues.

### Dual-tissue model for data integration

As in mammals, dietary nutrients ingested by *C. elegans* are not immediately available to all tissues. In *C. elegans*, the bacterial diet is first ingested and then ground by the pharynx to be delivered to the intestinal lumen. There, intestinal cells uptake bacterial biomass components, degrade macromolecules to extract nutrients, and

---

**Figure 1. Overview of the updated *Caenorhabditis elegans* metabolic network model and gene expression dataset used to derive tissue-relevant functions.** ▶

A  Computational pipeline to predict tissue function using tissue-level gene expression data.
B  Cartoon outlining the update of the *C. elegans* metabolic network model. GPR, gene-protein-reaction association.
C  Conceptual overview of integration of iCEL1314 with four categories of genes: highly, moderately, lowly, and rarely expressed. The predicted flux state in a tissue is a flux distribution that trails reactions associated with highly expressed genes in that tissue, while avoiding those associated with lowly expressed and rarely expressed genes. Circles and arrows indicate metabolites and reactions, respectively. Black arrows show flux, with thicker arrows indicating higher flux. Boxes depict enzymes encoded by genes that have expression levels indicated by color. Dashed arrows indicate reactions with no flux in the preliminary flux distribution stage according to Fig 2B but are then detected as latent reactions and are forced to carry flux when possible (see text for details).
D  To derive tissue-relevant metabolic network functions, a gene expression dataset obtained with single-cell RNA-seq of L2 animals was used (Cao *et al*, 2017). Single-cell data were combined by the authors to provide high-quality gene expression data for the seven tissues shown.
E  Distribution of metabolic genes in iCEL1314 in different expression categories in each individual tissue and in all tissues combined, with colors as in (B). For the combination of data, the union set of highly expressed genes and the intersection set of rarely and lowly expressed genes are illustrated with corresponding colors. One gene which was lowly expressed in some tissues and rarely expressed in others is not shown in the combined data.

**Figure 1.**

subsequently deliver transportable nutrients to other tissues. To properly simulate nutrient uptake, processing, and delivery, we developed a "dual-tissue model" that assumes nutrient exchange between the intestine and the six other tissues. This model has four compartments: (i) the intestine, which receives, metabolizes, and delivers nutrients to the other tissue; (ii) the intestinal lumen from which bacterial nutrients are imported by the intestine; (iii) the other tissue; and (iv) the extracellular space through which nutrients are exchanged between the intestine and the other tissue (Fig 2A, Table EV5; Appendix Table S1, Materials and Methods). Thus, the dual-tissue model allows the simulation of the metabolism of intestine and one other tissue at a time. We constrained this model such that the majority of nutrients used by *C. elegans* are obtained from bacteria, with limited use of "side nutrients". These mainly are importable molecules that may be present in the growth media or the bacterial diet but may be absent or quantitatively misrepresented in the assumed biomass composition of the bacteria (Yilmaz & Walhout, 2016) (Table EV3). We also allowed minimal usage of storage molecules including glycogen, triacylglycerides, and trehalose (Fig 2A, Materials and Methods).

We integrated the L2 tissue gene expression data with the dual-tissue model in two steps (Fig 2A). First, the gene expression data from each non-intestinal tissue was integrated with the other tissue compartment, one tissue at a time. Since the intestine can support the metabolism of the other tissue through the exchange of metabolites, leaving the intestine network free of any constraints may result in unrealistic predictions in this step, as enzymes not expressed in the intestine could then be used for metabolic conversions. We therefore discouraged flow-through reactions associated with genes that are lowly or rarely expressed in the intestine (Figs 1C and 2A). Second, the intestine gene expression data were integrated with the intestine compartment, while the overall nutritional exchange between the intestine and the other six tissues was imposed. This exchange was represented by transport fluxes that were calculated based on a combined flux distribution for the six non-intestinal tissues (Fig EV4A). Hence, the intestine not only has to adhere to its own gene expression levels, but also supply the cumulative nutritional requirement of other tissues (Fig 2A).

## Integration algorithm

Several algorithms are available to integrate metabolic network models with gene expression data (Lewis *et al*, 2010; Machado & Herrgard, 2014). However, most of these algorithms have at least one of the following limitations (Machado & Herrgard, 2014): (i) dependence on a single objective function such as the maximization of biomass production, (ii) dependence on pairwise comparisons with a reference state, or (iii) inability to produce a flux distribution where reaction directionality is addressed. We selected the iMAT algorithm (Shlomi *et al*, 2008; Zur *et al*, 2010) as a starting point for our network-level analysis, since this algorithm does not have any of the mentioned limitations and since it was specifically designed to integrate tissue-level data (Shlomi *et al*, 2008).

We optimized iMAT to what we refer to as iMAT++ to eliminate two drawbacks of the original algorithm. The first drawback relates to cases where highly expressed genes are associated with multiple reactions (e.g., gene *y* in the toy model in Fig 1C). The original iMAT approach is reaction-centered and tends to activate all reactions associated with highly expressed genes (Fig EV4B). However, the high expression of a gene may be indicative of only a subset of its associated reactions being active (e.g., only one reaction of the *y* gene, 6 → 7, carries flux in Fig 1C). To address this issue, we allowed the algorithm to choose a subset of reactions associated with a highly expressed gene and impose flux on only those. The second drawback of the original algorithm is the lack of a distinction between genes that are not expressed and those that are lowly expressed. iMAT is programmed to eliminate flux for any reaction that depends on genes below an arbitrary expression threshold. However, lowly expressed genes may be indicative of low flux rather than no flux (Yilmaz & Walhout, 2016). Therefore, instead of forcing reactions of lowly expressed genes to carry no flux, we aimed to minimize the sum of their fluxes (absolute), so that they may carry flux if needed by the rest of the network, but at the lowest possible level.

iMAT++ has two parts, and the flow of the first part is as follows: First, the number of highly expressed genes associated with at least one flux-carrying reaction and the number of no-flux reactions dependent on rarely expressed genes are summed ($Z_{fit}$) and

---

**Figure 2. Integration of iCEL1314 with tissue-relevant gene expression data.**

A Dual-tissue model used for compartmentalization of iCEL1314 during data integration. The two major compartments used are the intestine, which is the point of entry for bacterial nutrients, and another tissue. The lower panel shows the two main steps of integration. First, gene expression data for each tissue except the intestine is integrated with the model individually. Second, integrated flux distributions from the first step are combined using tissue weights that represent the relative mass and activity of each tissue (Fig EV4A, Appendix Supplementary Methods) and the intestine gene expression data is integrated.

B Flow chart of the optimized integration algorithm. A maximized or minimized variable from a step is carried to the next step as a constraint as shown by equations by the arrows (a bold uppercase term indicates a maximized or minimized sum of variables from the previous step). The δ term stands for small numbers that indicate the tolerance of deviation from the corresponding minimized flux sums. A latent reaction is a reaction that is only associated with highly expressed genes and converts metabolites that are available in the present state of the flux distribution, but does not carry any flux. See text and Appendix Supplementary Methods for details.

C Example pathways that share genes (only a relevant subset of reactions is shown for each pathway). Dashed arrows indicate skipped parts of the pathway and the rest of the metabolic network. Upper right panel shows expression categories of relevant genes in tissues. Lower right panel shows predicted flux in the propionate shunt obtained with iMAT and iMAT++ algorithms. Epsilon indicates the minimum flux imposed on reactions associated with highly expressed genes during integration (ε = 0.01 for every reaction shown).

D Analysis of agreement between experimental data and integrated flux distribution. The left panel shows percentage (*y*-axis) and number (bold numbers) of highly expressed genes that have no association with any flux-carrying reactions. The middle panel shows the same for reactions that depend on rarely expressed genes, but carry flux in the integrated network. The right panel shows the depletion rate of flux in lowly expressed reactions, which is calculated as one minus the ratio of total flux in these reactions to what is expected for the same number of flux-carrying reactions on average. In each panel, the results for exactly the same set of genes or reactions were extracted from the output of each algorithm and compared (Appendix Supplementary Methods).

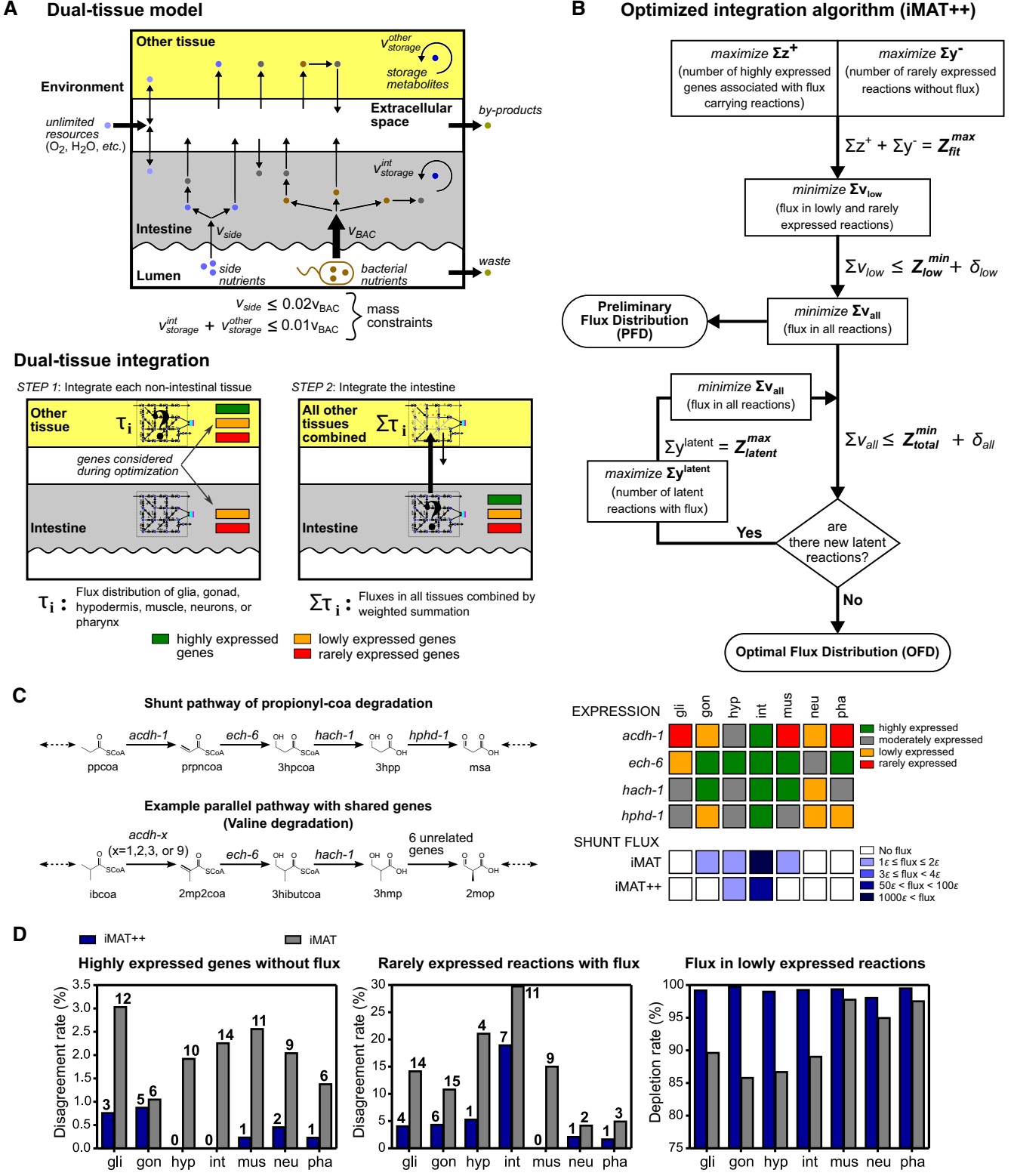

**Figure 2.**

simultaneously maximized to get $Z_{fit}^{max}$ (Fig 2B). Importantly, lowly expressed genes are excluded in this step and integrated into the next phase. Second, while $Z_{fit}^{max}$ is held constant, the total flux in reactions strictly dependent on lowly or rarely expressed genes is

minimized. This step covers rarely expressed genes to avoid increasing possible fluxes that may not have been fitted (forced to zero) during the first step. Then, while $Z_{fit}^{max}$ is held constant and the low flux sum is tightly constrained, the total flux in the entire

network is minimized based on a parsimonious FBA approach (pFBA) (Lewis *et al*, 2010; Machado & Herrgard, 2014; Yilmaz & Walhout, 2016). Together, these steps achieve a preliminary flux distribution (PFD) (Fig 2B, Table EV5).

The second part of iMAT++ deals with a compromise introduced by the gene-centered integration approach. A typical PFD has multiple "latent" reactions, which have no flux even though they are only associated with highly expressed genes and their reactants are available in the integrated network (e.g., 3 → 4 reaction associated with gene *x* in Fig 1C). These reactions exist because the associated genes are active somewhere else in the network and are therefore already integrated with the model (e.g., the flux in the 3 → 7 reaction addresses the high expression of the *x* gene in Fig 1C). We reasoned that latent reactions should carry flux, unless their products cannot be drained by other reactions. As a remedy, we identified all latent reactions and imposed flux on them (Fig 2B). Since the redistributed flux may create new latent reactions, this part of the algorithm is iterative. After latent reactions are addressed, an optimal flux distribution (OFD) is achieved (Fig 2B, Table EV5). For each tissue, the pertaining OFD represents the flux distribution that best fits the categorized gene expression data.

## Agreement between gene expression and optimal flux distributions

We verified the successful integration of tissue-level gene expression by iMAT++ by inspecting the flux predictions for well-understood pathways and by evaluating the overall fitting quality. As a specific example, flux predictions in a shunt pathway that degrades propionate or propionyl-CoA (ppcoa) (Watson *et al*, 2016) demonstrate the advantage of the gene-centered approach (Fig 2C). The *acdh-1* gene acts as a control point to allow the degradation of ppcoa by the shunt when the canonical ppcoa breakdown pathway is perturbed genetically or by low dietary vitamin B12 (Watson *et al*, 2016; Bulcha *et al*, 2019). This gene is highly expressed in the intestine, moderately expressed in the hypodermis, and lowly or not expressed in other tissues (Figs 2C and EV4C). Two of the other propionate shunt genes, *ech-6* and *hach-1*, are more broadly expressed. However, these genes are not only involved in the propionate shunt but are also associated with the breakdown of valine and isoleucine (Fig 2C). Therefore, tissue-level expression data suggest that the shunt pathway is active only in the intestine and hypodermis, consistent with direct assays (Arda *et al*, 2010; MacNeil *et al*, 2013). To compare the predictions from the two algorithms, the original, reaction-centered iMAT placed flux in the shunt pathway in four tissues, while iMAT++ more correctly restricted flux to the intestine and hypodermis (Fig 2C).

We evaluated the overall fitting quality based on three criteria, all of which revealed improvements in iMAT++ integrations compared with iMAT (Fig 2D). First, iMAT++ yielded very few highly expressed genes that were not associated with any flux-carrying reaction. Second, only few reactions that depend on rarely expressed genes carried flux. Third, the average flux in reactions dependent on lowly expressed genes was greatly depleted compared with the average flux in all flux-carrying reactions, thus indicating that the separate minimization step carried out for lowly expressed reactions worked effectively. In addition, the number of such reactions that carried flux was overall ~ 5% greater in iMAT than in

iMAT++, although iMAT is designed to minimize the number of these reactions instead of minimizing their total flux.

## Validation of predicted flux distributions based on known tissue functions

Next, we asked whether OFDs captured tissue-level metabolic functions consistent with our current knowledge of *C. elegans* physiology. We generated a heat map of reactions that have enriched or depleted fluxes in one or two tissues, or that have uniform flux profiles across all tissues (Fig 3A, Table EV6). We found one common and five tissue-specific clusters of reactions that contained metabolic functions consistent with their tissue pattern and current knowledge. The common cluster (Fig 3A) includes many reactions that either produce biomass precursors or assemble biomass from these precursors, while some are part of energy production. Thus, all tissues are predicted to produce biomass, consistent with the fact that the body size of *C. elegans* increases dramatically as it proceeds through the different stages of larval development (Hirsh *et al*, 1976). Moreover, biomass production alone consumes a significant portion of assimilated nutrients as shown with biomass yield calculations (Fig 3B), which shows that OFDs captured the metabolic burden of growth in growing larvae (Ferris *et al*, 1995; Yilmaz & Walhout, 2016).

Other consistent tissue-level predictions included (Fig 3A) the degradation of bacterial macromolecules in the intestine, a relatively large flux for the propionate shunt in the intestine (see also Fig 2C) (MacNeil *et al*, 2015; Watson *et al*, 2016; Bulcha *et al*, 2019), unique fatty acid processes in the intestine consistent with the presence of lipid droplets and production of yolk in this tissue (Hall *et al*, 1999; Lemieux & Ashrafi, 2015; Vrablik *et al*, 2015), ascaroside production in the intestine and hypodermis (Park & Paik, 2017; Artyukhin *et al*, 2018), the production and secretion of the neurotransmitters octopamine and serotonin in neurons, and the biosynthesis of DNA in the hypodermis and gonad. The latter prediction suggests that the hypodermis and gonad are the major tissues where significant cell proliferation occurs, which agrees with the cell lineage where the hypodermis and gonad are the tissues with the largest fraction of cells dividing at the early L2 stage (Sulston *et al*, 1983; Hubbard & Greenstein, 2005), and with the fact that the gonad gene expression data are dominated by constantly dividing germline cells (Cao *et al*, 2017). In addition, a hypodermis-specific group of reactions includes functions shared with mammalian skin or liver, such as urocanic acid production (Gibbs & Norval, 2011) and degradation (Kalafatic *et al*, 1980), kynurenine (Claria *et al*, 2019) and cysteine (Stipanuk *et al*, 2006) metabolism, and lysine degradation (Papes *et al*, 1999) (Fig 3A). The hypodermis functions as the skin of *C. elegans*, based on its physiological role and structure (Chisholm & Xu, 2012). However, it has recently been shown that the hypodermis transcriptome from adult *C. elegans* is best correlated with that of human liver (Kaletsky *et al*, 2018), and this relationship is also captured by our hypodermis model. Taken together, these results show that tissue-level integration of gene expression with iCEL1314 using iMAT++ captures biologically relevant functions.

## Flux variability analysis of tissue-level metabolic network models

While OFDs represent flux distributions that optimally fit categorized tissue gene expression data, they do not serve as unique

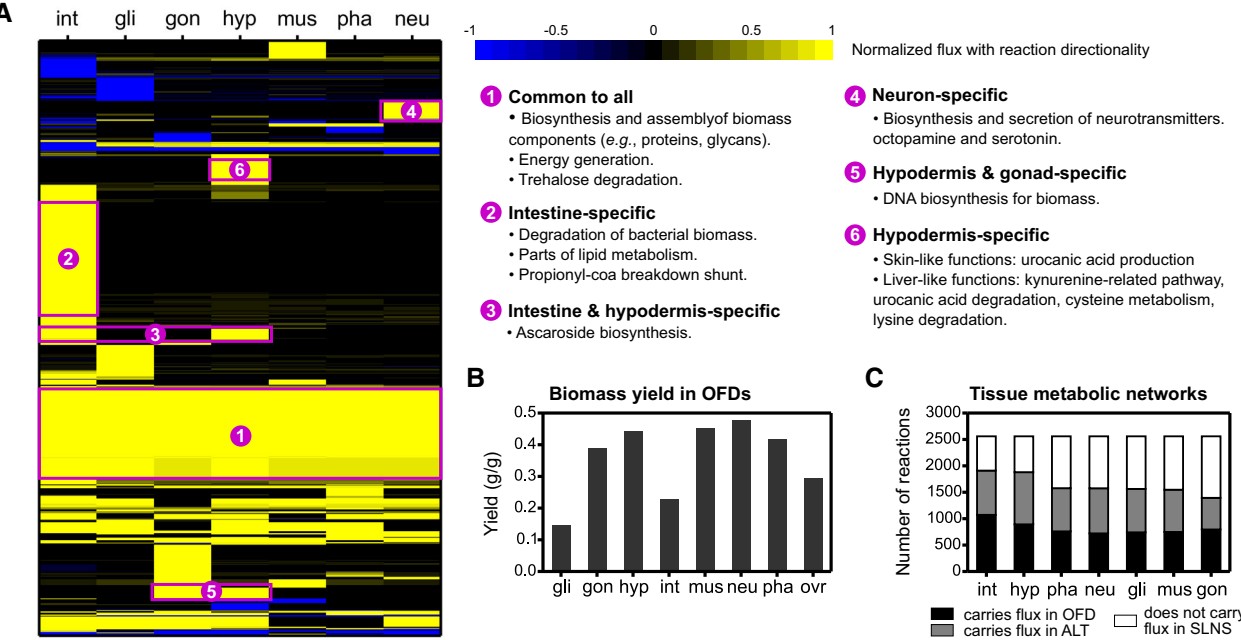

**Figure 3. Validation of OFDs based on known tissue functions.**

A  Heat map of reaction fluxes (rows normalized by the absolute maximum) for a set of 927 reactions (Table EV6) that are biased in flux profiles toward enrichment or depletion in one or two tissues, or that have flux in all tissues. Functional observations associated with six clusters are indicated on the right. The analysis distinguishes between flux and no flux for any reaction, as well as between positive and negative flux values for reversible reactions, which correspond to a flux in forward and reverse directions, respectively.

B  Biomass yield of each tissue based on optimal flux distributions (OFDs), defined as grams of biomass produced divided by grams of substrates (nutrients) consumed. The abbreviation ovr stands for overall biomass yield, which was calculated by combining intestine biomass with a weighted sum of other tissue biomass, and using the nutrient input from the intestine integration step (Fig 2A) (see Appendix Supplementary Methods for a detailed description of calculations).

C  Tissue metabolic network breakdown based on flux variability analysis. Reactions in the model were converted to single-direction reactions by dividing reversible reactions into forward and reverse reactions. Each single-direction reaction was categorized for each tissue as carrying flux in the optimal flux distribution (OFD), carrying flux in alternate flux distributions (ALT), or not carrying flux in the feasible solution space (SLNS) of iMAT++ integrations. For each tissue, the sum of the number of OFD and ALT reactions indicate the size of the accessible network.

solutions. The flux states of many reactions in an OFD can be changed from carrying no flux to carrying flux, or vice versa, without deteriorating data-fitting quality of iMAT++ (Fig 2B and D), except for a small change in total flux. For instance, the ascaroside production pathway in the intestine OFD produces and secretes only ascr#2. However, coA forms of five other ascarosides are synthesized prior to the synthesis of ascr#2 (Fig EV1), and these ascarosides can also be made and secreted from the intestine without affecting the agreement of the flux distribution with highly, lowly, and rarely expressed genes. The reason this is not reflected in the intestine OFD is because an OFD is a single solution that explains gene expression categories with minimum total flux. Thus, we can infer that ascr#3 production is in alternate optimal flux distributions (ALT) (Orth *et al*, 2010) for the intestine network. To systematically capture all such metabolic functions in ALT of each tissue, we performed flux variability analysis (FVA) (Mahadevan *et al*, 2002). This analysis yielded minimum and maximum flux each reaction can take (Table EV5), while fitting constraints generated by the end of the iMAT++ optimization are maintained (Fig 2B). We then categorized reactions in each tissue as those carrying flux in OFD, those carrying flux in ALT, and those not carrying flux in the feasible solution space (Fig 3C). Reversible reactions were evaluated in each direction (forward and reverse) separately. We defined a tissue-level

metabolic network (Fig 1A) based on the combined set of reactions in OFD and ALT, as these reactions are all accessible in optimal solutions. Interestingly, the intestine and hypodermis were found to have the largest metabolic networks (Fig 3C), confirming the metabolic role of these two tissues.

**Flux potential analysis for quantitative integration of expression data**

The semi-quantitative integration of RNA-seq data followed by FVA yields information about tissue metabolic networks at the network level but is not designed to capture quantitative gene expression values. An example for a good capture of differential expression is the flux prediction for the first reaction in the propionate shunt, which is catalyzed by ACDH-1 (Watson *et al*, 2016) (Figs 2C and 4A). In many cases, however, the integration algorithm does not yield reaction fluxes that correlate well with the expression levels of the associated genes. For example, *ldh-1* is highly, but differentially, expressed in four tissues, but each of these tissues has the same predicted flux in the LDH reaction, and therefore is predicted to produce the same amount of lactate (Fig 4A). We manually evaluated the lactate production potential of tissues by analyzing the expression profiles of key genes in the pathway that converts

## A

### Expression-flux relationships

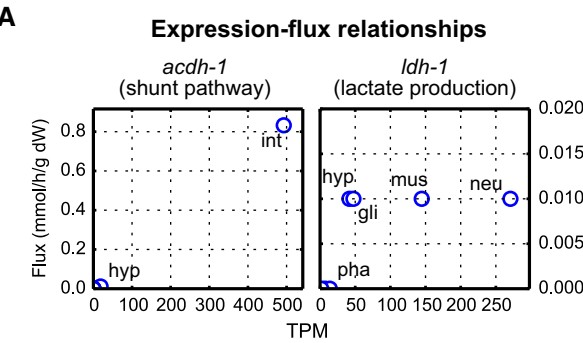

## B

### Lactate production pathway

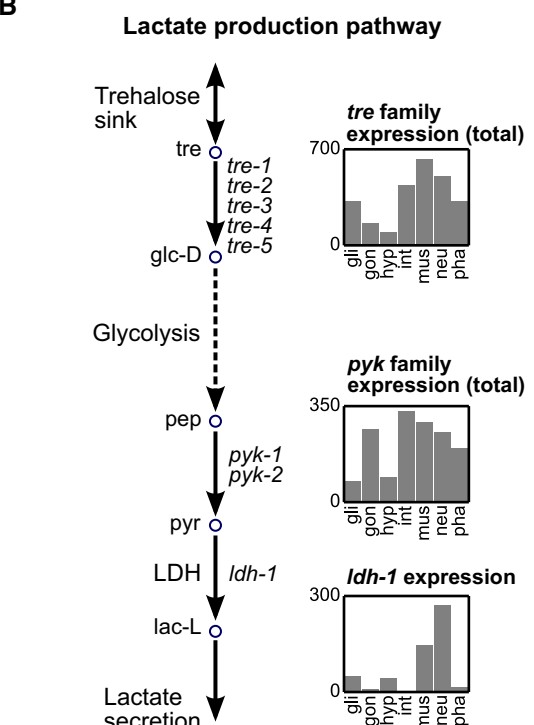

**Figure 4. The need for quantitative assessments.**

A Example relationships between gene expression levels and flux for the indicated gene/pathway pairs.

B Lactate production pathway reconstructed based on Ref. (Braeckman *et al*, 2009; Watts & Ristow, 2017). Bar charts show gene expression profiles (in TPM). When multiple genes are connected by "or" in reaction association (e.g., paralogs), as in the case of *pyk* and *tre* families, total expression is used. LDH, lactate dehydrogenase.

trehalose to lactate, assuming this is the main route for lactate production (Braeckman *et al*, 2009; Watts & Ristow, 2017) (Fig 4B). We deduced that neurons and muscles have a greater lactate production potential than others, not only because of *ldh-1* levels alone, but also since genes associated with other key reactions of the pathway are highly expressed in these tissues (Fig 4B). Due to the integration of gene expression data using four discrete categories of gene expression levels, tissue-relevant OFDs are generally not capable of delivering quantitative predictions. However, it is also not feasible to make quantitative predictions for every reaction by

manually mapping expression data to the metabolic network. This would be a difficult and time-consuming task especially because the fluxes of surrounding reactions need to be taken into account, and not all reactions are part of a previously studied pathway as in the case of lactate production.

To quantitatively integrate gene expression differences at the pathway level, we developed a method that we refer to as flux potential analysis (FPA). We describe the concept of FPA by a toy network (Fig 5A). FPA uses the entire network of reactions after reversible reactions are converted to irreversible forward and reverse reactions but focuses on one "target" reaction at a time (reaction $k$, Fig 5A). The flux potential of the target reaction ($FP_k$) is defined for each tissue (T) as a function of three sets of variables: (i) the reaction network of the tissue defined as reactions in OFD and ALT (Fig 3C), (ii) expression levels of genes associated with reactions in this network relative to other tissues, and (iii) the metabolic distance (Materials and Methods) of every reaction of this network from the target reaction. The distance factor is included to promote pathway-level integration, with the reasoning that more distant parts of the metabolic network (as in other pathways) will have lesser influence on the substrates of the target reaction. For $FP_k$ in a tissue to have a contextual meaning, it is normalized by $FP_k$ for a hypothetical "super" system (S), wherein all metabolic genes are expressed at the maximum level found in any tissue. Thus, FPA of the target reaction yields a dimensionless value between 0 and 1 for each tissue, which is named relative flux potential ($rFP_k$) (Fig 5A).

To calculate $FP_k$, we maximized the flux of the target reaction such that a weighted sum of all fluxes is limited by an arbitrary number called flux allowance ($a$) (Fig 5A). The weight of each reaction in the tissue network acts as a penalty for placing flux in that reaction, as a larger weight indicates more allowance spent per unit flux, and therefore, less allowance left for the flux of the target reaction, which is to be maximized. Reaction weights are made inversely proportional to the normalized expression of associated genes, so that tissues with lower expression are penalized more. The weights are also made inversely proportional to metabolic distance (increased by 1 to avoid infinity), so that the flux in reactions more distant to the target reaction have less influence on $FP_k$. The distance is raised to a power ($n$) called distance order, which controls how fast the influence of remote reactions decays with distance. In the toy example (Fig 5A), assuming a distance order of 1, the first tissue has a larger $rFP_k$ than the other two, thanks to lower weights in this tissue for two reactions proximal to the target reaction. The third tissue has zero flux potential as the relevant pathway is not part of its network.

We used FPA to evaluate lactate production potential of each of the seven tissues. First, we targeted the LDH reaction (Fig 4B), which converts pyruvate into lactate (Fig 5B). Since we did not initially have a trained distance order to use, we calculated flux potentials by varying this parameter (Fig 5B). At high distance orders ($\geq 6$), $rFP$ values converged to relative expression levels of *ldh-1* (i.e., the weight of the target reaction), as the weights, and hence the influence, of other reactions decayed to zero. At lower distance orders, the contribution of other reactions in the pathway became evident, and the difference between LDH flux potential of muscle and neurons was reduced. Importantly, FPA used the presumed pathway (Fig 4B) (Braeckman *et al*, 2009; Watts & Ristow, 2017) to

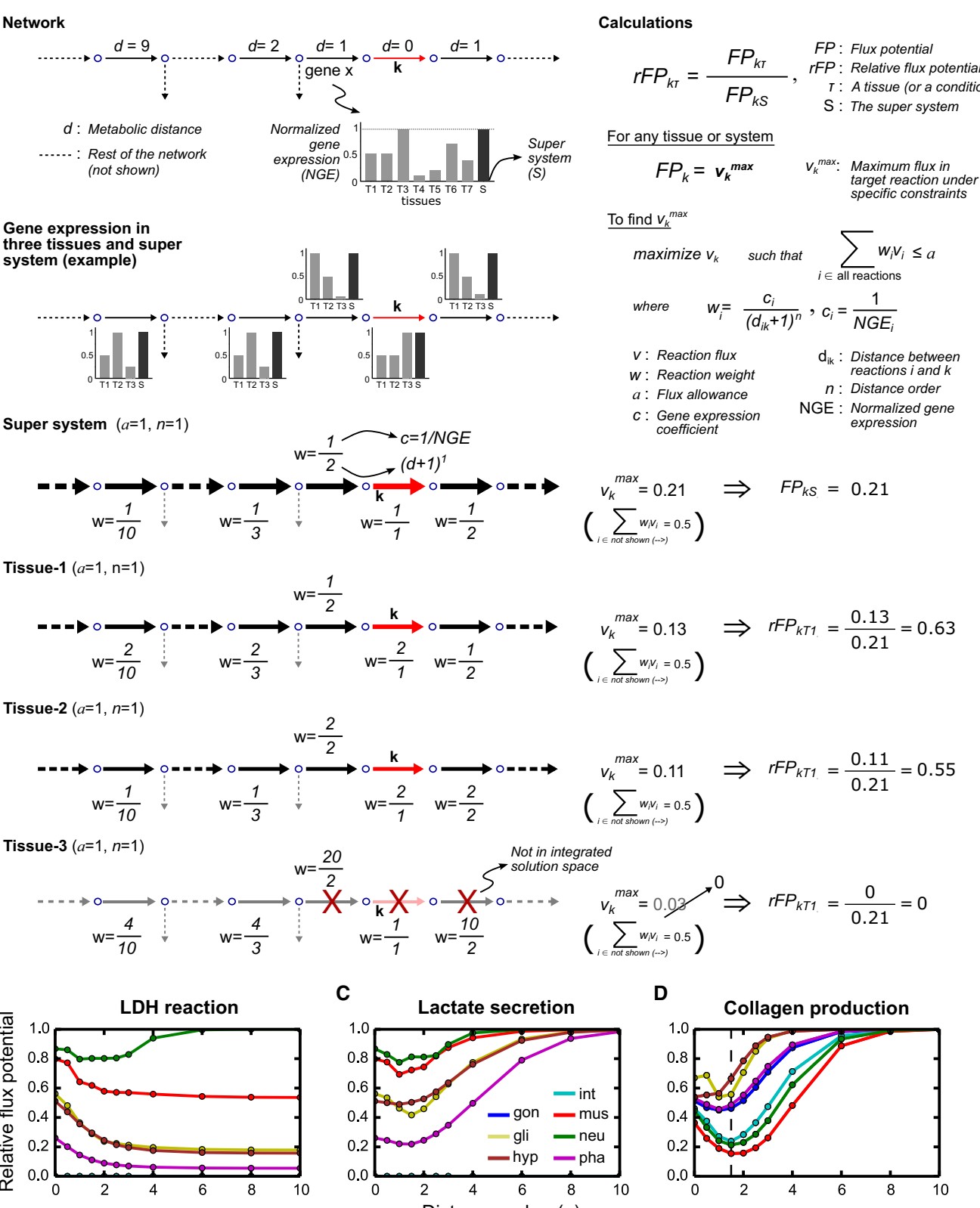

## A  Flux Potential Analysis (FPA)

**Figure 5.**

**Figure 5. Flux potential analysis.**

A  Flux potential analysis explained with a toy network and three hypothetical tissues that are a subset of a dataset with seven tissues total. The analysis is carried out for a target reaction (*k*) shown in red. Flux potential of the target reaction (*FP*) is calculated in each tissue using FBA. Flux of target reaction is maximized such that a weighted sum of fluxes (all fluxes are positive values since reversible reactions are divided into two reactions so that every reaction has a single direction) is constrained by an allowed total flux (*a*). The formulation of this constraint and reaction weights (*ω*) makes flux potential a function of normalized gene expression (NGE, bar charts) and metabolic distance (*d*, top panel) over the network. The theoretical maximum of flux potential is calculated using the super system (*S*), a hypothetical tissue that expresses every metabolic gene at the highest level of the seven tissues. To calculate relative flux potential (*rFP*), flux potentials from individual tissues are divided by this theoretical maximum. Dashed arrows indicate multiple reactions that are not shown for simplicity. During flux potential calculations, the contribution of these reactions to the weighted sum is assumed to be 0.5 in every system, as shown in parentheses. The total allowance was taken as 1.0. All fluxes in the reactions shown must be the same due to mass balance laws. Therefore, the calculated flux potentials are equal to 1.0–0.5 divided by total weight in each condition. A distance order of 1 (*n* = 1) is used for weight calculations. Red cross in Tissue 3 calculation indicates the elimination of a reaction during FBA based on prior FVA analysis (i.e., reaction not part of OFD or ALT solutions). Deleted flux potential value (0.03) indicates what would be found for Tissue 3 if these reactions were not eliminated.

B–D  Flux potential of reversed LDH reaction (B), lactate export (C), and collagen demand reaction (D) as a function of distance order. Dashed line (D) indicates order of choice for subsequent analyses.

activate the LDH reaction, which demonstrates the successful integration of pathway-level information.

Flux potential analysis can also be used at metabolite level by focusing on a reaction that drains or introduces a metabolite of interest, such as lactate. The flux potential of a target reaction that drains this metabolite shows its production potential. Likewise, the flux potential of one that introduces the metabolite of interest shows the degradation potential for it. We evaluated lactate production potential at metabolite level by calculating the *rFP* of the reaction that exports lactate (Figs 4B and 5C). At lower distance orders, the results showed similar differences as in LDH analysis (Fig 5B) as LDH is the only reaction that produces lactate. At higher distance orders, all tissues converged to a relative flux potential of 1, reflecting the fact that there is no gene association for lactate transport, and therefore, the flux penalty for the target reaction itself is uniformly 1 in all tissues. The lack of a gene expression value for the target reaction is a general property of metabolite-level FPA, as most transporters are unknown and demand and sink reactions have no GPR associations by default (Yilmaz & Walhout, 2016). This property agrees with our goal of metabolite-level analyses, which is to evaluate an overall metabolite production or degradation potential considering all reactions and pathways that may produce or consume the metabolite of interest.

An appropriate distance order value is needed for the systematic application of FPA to all reactions and terminal metabolites in the tissue network models. To optimize the distance order, we analyzed flux potential of collagen production in different tissues. We chose collagen production because collagen is both a metabolite in the network and also a protein whose expression can be approximated by using the scRNA-seq data using *C. elegans* collagen-encoding genes. Furthermore, the hypodermis is known as the main site of collagen biosynthesis in *C. elegans*, to support cuticle production (Johnstone, 1994). Indeed, when we inspected the gene expression levels of all annotated collagen genes, we found that the hypodermis expresses most of these genes at the highest level, with glia being a secondary site (Appendix Fig S4A). To enable metabolite-level analysis, we first added a collagen demand reaction to the model (Appendix Fig S4B). Flux potential calculations for this reaction showed results consistent with our expectations at distance orders 1 through 2.5, with 1.5 showing the largest difference between hypodermis and glia (Fig 5D). Based on this result and visual inspection of other data (e.g., Fig 5A and B), we selected a distance order of 1.5 for subsequent FPA.

**Systematic analysis of tissue function based on flux potentials**

Next, we calculated relative flux potentials for all reactions. We combined this data with OFD and FVA results and divided into reaction-level (i.e., relative flux potential of regular reactions, Table EV7) and metabolite-level (transport, demand, and sink reactions, Table EV8) predictions. To derive tissue-enriched metabolic functions from these datasets, we extracted reactions that showed significant variation of flux potential among tissues and were biased toward one or two tissues as the sites with the highest potential. For such reactions, we call the top two tissues in relative flux potential as primary and secondary sites (Tables EV7 and EV8, Appendix Supplementary Methods).

Out of 1,647 reactions in the reaction-level dataset (Table EV7), 1,114 were tissue-enriched (i.e., assigned to a primary or secondary tissues) based on *rFP* differentials. An important question to address is whether an FPA-based prediction is consistent with the network-level flux distributions. When a reaction carries flux in the OFD of a primary or secondary tissue where it was found to have a high flux potential, we have a higher confidence in our prediction as both network- and reaction-level analyses agree. Thus, we also separated out tissue-enriched reactions that carried flux in OFD of the corresponding primary or secondary tissues, which narrowed down predictions to 790 tissue-specific reactions (Fig 6A, starred primary and secondary sites in Table EV7). Other tissue-specific reactions were in the ALT of the primary or secondary tissues and are associated with a lower confidence. Reaction-level FPA (Fig 6A) revealed many differential tissue functions missed by the semi-quantitative approaches. One example includes the relatively large potential of some reactions in fatty acid beta-oxidation, TCA cycle, and electron transport chain of muscle, which is consistent with the energetic requirements of this tissue (Barclay, 2017; Laranjeiro *et al*, 2017). Another example showed that neurons have a relatively large potential in the metabolism of cyclic AMP and GMP, as well as phosphatidylinositols (Fig 6A). In the mammalian nervous system, cAMP and cGMP are key intracellular metabolites for signal transduction (Gorshkov & Zhang, 2014), and different forms of phosphatidylinositols play important roles in synaptic transmission (Frere *et al*, 2012). In addition to these new predictions, we found that tissue-relevant functions predicted by semi-quantitative integration (Fig 3A) were also captured by relative flux potentials (Fig 6A).

The metabolite-level FPA dataset included 396 metabolites (Table EV8), of which 250 were associated with tissue-specific

production ($N$ = 138), degradation ($N$ = 50), or both ($N$ = 62) based on high-confidence predictions (Appendix Fig S2). As different from the reaction-level analysis, we labeled the tissue-enriched flux potential predictions for a metabolite with high confidence when the pertaining OFD showed zero flux for both the degradation and production of the metabolite in the primary or secondary tissues. This is because the flux minimization step of iMAT++ (Fig 2B) minimizes metabolite input and output through transport, demand, and sink reactions, and therefore, a lack of flux in these reactions does not indicate a real prediction, as was exemplified with the production of different ascarosides above.

Assignments of metabolite production and degradation to primary and secondary tissues (Table EV8) captured many biologically relevant metabolite–tissue associations. For example, multiple metabolites produced by the mammalian nervous system were also predicted to be produced in neurons and glia in *C. elegans* (Fig 6B). We accurately predicted additional neurotransmitters to be produced in neurons, including acetylcholine (ach in Fig 6B), 4-aminobutanoate (GABA, 4abut), beta-alanine (ala-B), dopamine (dopa), and histamine (hista). Neurons were also the primary site for the degradation of amino acids aspartate (asp-L) and glutamine (gln-L; Fig 6C), owing to the efficient conversion of these metabolites to ala-B and GABA, respectively. Taurine (taur) is also predicted to be primarily produced by neurons (Fig 6B) and degraded by glia (Fig 6C) to form glutaurine (glutaur, Fig 6B). Taurine plays multiple roles in the nervous system as a neurotransmitter, neuromodulator, and neuroprotectant, as well as an osmolyte (Wu & Prentice, 2010; Ripps & Shen, 2012). Glutaurine is a peptide found in the brain (Bittner *et al*, 2005). Glia are also predicted to be a primary site for production of multiple purine and pyrimidine compounds (Fig 6B) including adenine (ade), adenosine (adn), guanine (gua), hypoxanthine (hxan), and deoxycytidine (dcytd). Both purines and pyrimidines are important in neuronal development (Fumagalli *et al*, 2017), and purines can serve as intercellular signaling molecules between neurons and glial cells (Fields & Burnstock, 2006) and as trophic substances in the same tissues (Rathbone *et al*, 1999). These results show that predicted tissue-level production and degradation potentials for metabolites can match *C. elegans* neurons and glia to the mammalian nervous system.

In the muscle, the predicted production of 3-amino-isobutyrate (BAIBA, 3aib) is consistent with this molecule being secreted to blood during exercise in humans (Roberts *et al*, 2014). Similarly, methylnicotinamide (1mncam) is produced by human skeletal muscle during adjustment to differences in an exercise regime (Strom *et al*, 2018) and is predicted to be produced in both muscle and pharynx (which includes pharyngeal muscle) of *C. elegans* (Fig 6B). Muscle is also predicted to be the primary degrader of energy-rich metabolites including fatty acids, branched-chain amino acids, trehalose, and glycogen (Fig 6C). Interestingly, we predict that ketone bodies beta-hydroxybutyrate (S3hb) and acetoacetate (acac) can also be most efficiently degraded by the muscle (Fig 6C). Thus, both reaction-level (Fig 6A) and metabolite-level (Fig 6C) analyses identified muscle as a major site that harvests the reducing power of various metabolites to generate energy.

We also found additional metabolite–tissue associations relevant to *C. elegans* physiology such as the production of eumelanin, representative of the cuticle melanin (Calvo *et al*, 2008), in hypodermis;

and the production of chitin, a building block of the grinder of pharynx and egg shells (Zhang *et al*, 2005; Straud *et al*, 2013; Stein & Golden, 2018), in pharynx and gonad (Fig 6B). The pharynx also has the highest flux potential for the chitinase reaction (RC01206f, Table EV7), potentially indicative of chitin turnover during the development and maintenance of the grinder. Finally, metabolite production predictions based on semi-quantitative integration were reproduced by FPA (e.g., octopamine, ascarosides, and urea; Figs 3A and 6B, Table EV8), with the ascaroside production capacities of intestine and hypodermis revealed for all ascarosides in the model (Figs 6B and EV1).

## Summary of findings with MERGE

Overall, we divide the predictions from the analysis of *C. elegans* metabolism with MERGE (Fig 1A) into three categories (Fig 7). The first category includes functions that were already "known" based on *C. elegans* physiology. The second one covers "coherent" predictions, which are consistent with mammalian physiology. The third category includes "unexplored" predictions that cannot be immediately verified. The known and coherent predictions validate the ability of our pipeline to capture biologically relevant functions, and novel predictions serve as future hypotheses.

In Fig 7, we exemplify some attractive novel predictions based on what we already know. For example, glia are the secondary site for eumelanin production (Tables EV7 and EV8, Fig 6B), which may be related to the presence of neuromelanin in glia and neurons in mammals (McCloskey *et al*, 1976; Hopley *et al*, 2017). The gonad is predicted to be the main selenocompound producer, including selenoprotein biosynthesis, which can be potentially linked to the essentiality of selenium in human sperm cells (Hawkes & Turek, 2001). In agreement with a recent study, we find that the hypodermis may act as liver in *C. elegans* based on distinct processes predicted to occur in this tissue, such as kynurenine metabolism (Kaletsky *et al*, 2018). The intestine is predicted to be unique for the processing of some lipids such as saturated medium-chain length fatty acids, which seems to be related to fat storage and yolk production functions of this tissue (Hall *et al*, 1999; Lemieux & Ashrafi, 2015; Vrablik *et al*, 2015). Muscle and neurons are predicted to be the main sites of bicarbonate production, which is consistent with the predicted lactate production in these tissues since bicarbonate acts as a buffer against acid accumulation (Beaver *et al*, 1986). Finally, the pharynx can efficiently make pyroglutamate (5oxpro) which may be due to the presence of pyroglutamate residues in the N-terminal of thyrotropin-releasing hormone peptides found in the pharynx (Van Sinay *et al*, 2017). Taken together, our predictions provide a rich resource to gain deeper insight into tissue metabolism.

## Robustness and usability of MERGE

MERGE is composed of three modules (Fig 1A) that use multiple parameters. To evaluate the robustness of our results to changes in variable parameters and methods, we performed sensitivity analyses at each module. Specifically, we changed a key parameter or a method in the pipeline and redid the entire analysis (Fig EV5, Appendix Supplementary Methods). First, we checked the sensitivity of our results to flux thresholds, arbitrary parameters used in

## A    Regular reactions

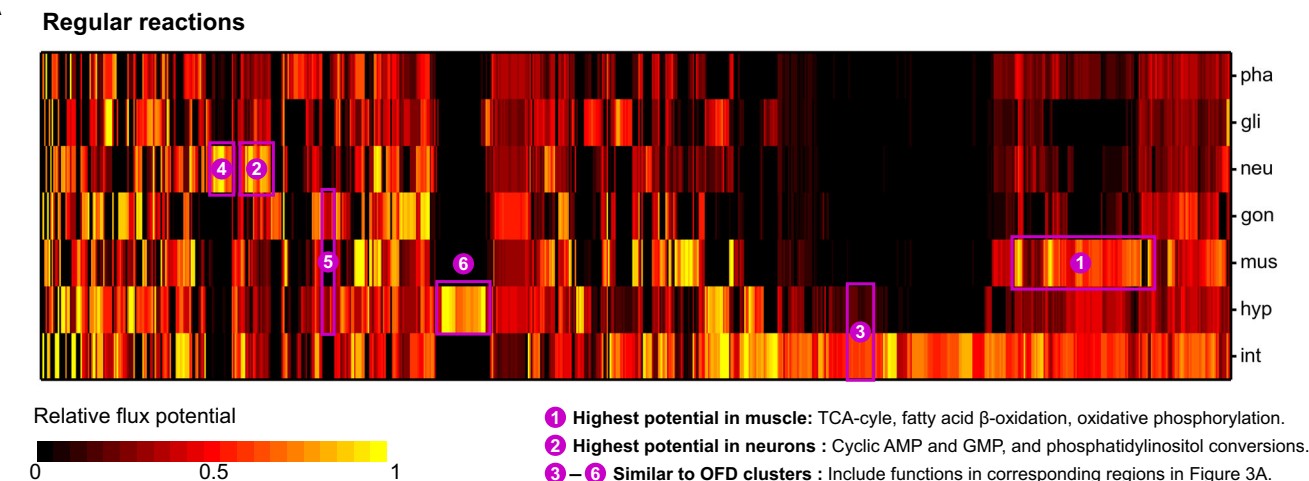

Relative flux potential

0          0.5          1

**①** **Highest potential in muscle:** TCA-cyle, fatty acid β-oxidation, oxidative phosphorylation.

**②** **Highest potential in neurons :** Cyclic AMP and GMP, and phosphatidylinositol conversions.

**③ – ⑥** **Similar to OFD clusters :** Include functions in corresponding regions in Figure 3A.

## B    Selected relative production potentials

## C    Selected relative degradation potentials

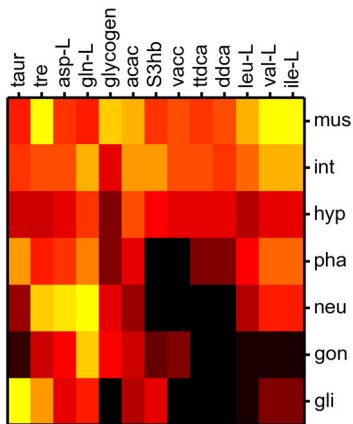

**Figure 6.   Systematic analysis of tissue function based on flux potentials.**

A    Heat map of relative flux potentials for regular metabolic model reactions, which exclude transports, exchanges, and demand/sink reactions. A subset of 1,114 reactions that yield tissue-specific flux potentials with high confidence is shown (see text and Appendix Supplementary Methods).

B    Production potential analysis of selected metabolites (extracted from Appendix Fig S2A) that are predicted to be produced at relatively high levels in one or two tissues with good confidence (see text and Appendix Supplementary Methods). Production potential is calculated based on an export or demand reaction that drains the metabolite.

C    Degradation potential analysis of selected metabolites (extracted from Appendix Fig S2B) that are predicted to be degraded relatively more efficiently in one or two tissues with good confidence (see text and Appendix Supplementary Methods). Degradation potential is calculated based on an import or sink reaction that introduces the metabolite.

iMAT++ that define significant flux. We found that changing flux thresholds in iMAT++ caused mostly numerical differences and did not affect high-confidence predictions (Fig 7, Tables EV7 and EV8) significantly (Fig EV4).

We next tried different FVA methods to build metabolic networks for FPA (Fig 1A). Network building is the rate-limiting step of MERGE and may cause usability problems with larger models such as human metabolic networks (Swainston *et al*, 2016; Brunk *et al*, 2018). Interestingly, skipping FVA and applying FPA on the entire network, as if every reaction can carry flux, did not change our key results in Fig 7 considerably (Fig EV5A), but did have an overall impact on the high-confidence predictions (Fig EV5B). To improve

the computational performance of the network building step with minimal compromise, we developed two modified versions of FVA (Appendix Supplementary Methods), both of which recapitulated all key findings (Fig EV5A) and had little impact on the entire set of high-confidence predictions (Fig EV5B). Thus, together with the original FVA, we provided three versions of the network building module with three speed levels, which the user can select depending on the model used and the computational resources available. With these modifications, MERGE should be applicable to most, if not all, complex models, including humans (see below).

The last module of the MERGE pipeline is FPA, which is a heuristic approach that uses a distance order to infer the contribution of

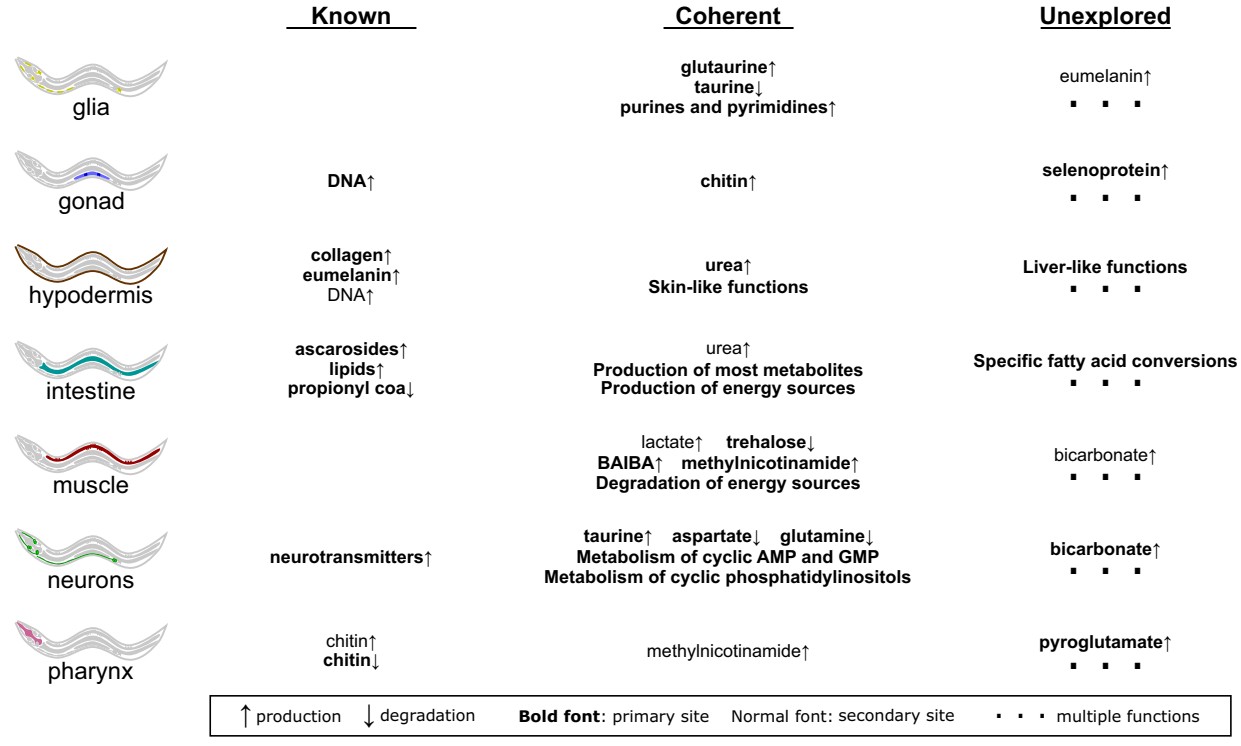

**Figure 7. Summary of integration analyses.**

Summary of functional predictions in Tables EV7 and EV8.

neighboring reactions to reaction flux potentials. Although our main results were largely robust to small changes in distance order (1–2), extreme values (0 and 10) had a larger impact resulting in the loss of some of the key predictions (Fig EV5). In this study, we found that a distance order of 1.5 was optimal to get verifiable predictions. For future applications of MERGE, we recommend careful tuning of the distance order for the optimization of predictive power.

## Discussion

In this study, we have integrated the *C. elegans* metabolic network model (Yilmaz & Walhout, 2016) with high-quality gene expression data (Cao *et al*, 2017) to provide insights into metabolism at the level of seven individual tissues. This work provides system-level insights into tissue-relevant metabolism that was heretofore not feasible. Our knowledge of tissue-relevant metabolism in *C. elegans* was limited because it is difficult to isolate individual tissues and monitor metabolic activity using metabolomics. To predict tissue-relevant metabolism in *C. elegans*, we developed MERGE, a novel computational pipeline that integrates genome-scale metabolic network models with gene expression data both qualitatively and quantitatively.

Parts of MERGE build on previously developed computational tools, and other parts are novel. For the first step of MERGE, we selected the iMAT algorithm (Shlomi *et al*, 2008) and modified it to what we refer to as iMAT++. This first step achieves an integration between the metabolic network model and gene expression data at the whole network scale, because a global flux distribution is fitted

to the entire metabolic network model. The iMAT++ adaptation uses a gene-centered rather than reaction-centered approach for the assignment of reaction fluxes based on gene expression data. As a result, iMAT++ is tailored to better agree with gene expression data. The second step of MERGE uses FVA (Mahadevan *et al*, 2002) to explore the fitting space at every model reaction and excludes reactions that do not carry flux in this space from the network. Finally, the third step of MERGE, FPA, is a new method that can be used to quantitatively predict reaction fluxes in the context of individual pathways and can also be set up to predict the production and consumption potentials of individual metabolites that are reactants in individual reactions. The latter was particularly useful for deriving tissue-specific functions (Figs 6 and 7). After applying FPA, we derived a set of high-confidence predictions by combining reaction flux potentials with the globally fitted flux distributions from the first and second step of MERGE (Fig 7, Tables EV6 and EV7). Importantly, most predictions were specifically derived after FPA, which indicates that this novel step is the most important component of MERGE with respect to predictive power. Indeed, if we stopped our analysis at iMAT++ level, we would have missed most of our predictions (Figs 3 and 7, Tables EV7 and EV8).

We show the application of MERGE to the prediction of tissue-relevant metabolism in *C. elegans* using a high-quality dataset based on scRNA-seq with L2 animals (Cao *et al*, 2017). Longer term, it will be important to extend MERGE to predictions for all life stages of the animal, to determine how tissue-relevant metabolism changes during development. In principle, MERGE should be applicable not only to tissue-relevant gene expression data, but to other types of

gene expression data as well, including different conditions, developmental stages, and genetic perturbations.

MERGE is not specific for use with *C. elegans* but should be broadly applicable to any system for which genome-scale metabolic network models and high-quality gene expression data are available. As a proof of principle to demonstrate the broad use of MERGE, we integrated the human metabolic model Recon 2.2 (Swainston *et al*, 2016) with transcriptomes of 17 tissues from a publicly available dataset (Uhlen *et al*, 2015). While applications to other organisms are beyond the scope of this study, we provide the code and results for this preliminary analysis in our repository (see below). The demonstration of applicability to humans shows easily verifiable predictions including, but not limited to, a relatively high flux potential of TCA cycle reactions in skeletal muscle, unique lipid metabolism in the digestive organs, and melanin production

potential in the skin, which are all in concordance with our predictions in the respective corresponding tissues in *C. elegans*.

In our study, we used mRNA levels as a proxy for enzyme activity. Although this led to numerous highly valid predictions, it is likely that this approach will not fully capture all tissue-relevant metabolism. This is because mRNA levels may not faithfully represent protein levels, and protein levels may not faithfully represent enzyme activity. For instance, allosteric mechanism by which metabolites directly regulate enzyme activity will be missed. In the future, it should be feasible to use MERGE with proteomic datasets.

MERGE is readily available and can be used in conjunction with the COBRA toolbox for metabolic network modeling (Heirendt *et al*, 2019). Finally, in the longer term, scRNA-seq datasets may produce sufficiently accurate expression data to model metabolic differentiation of individual cells.

# Materials and Methods

**Reagents and Tools table**

| Reagent/Resource | Reference or source | Identifier or catalog number |
|---|---|---|
| **Experimental Models** | | |
| Cell types representing seven major tissues (*C. elegans*) | Cao *et al* (2017) | |
| **Software** | | |
| Python 2.7 | https://www.python.org | |
| MATLAB 2019a | https://www.mathworks.com | |
| Gurobi Optimizer 8[a] | https://www.gurobi.com/ | |
| **Other** | | |
| iCEL1314 (genome-scale metabolic network model of *C. elegans*) | This study, Yilmaz & Walhout (2016) | BioModels (Chelliah *et al*, 2015): MODEL2007280001 |

[a] See Appendix Table S2 for solver parameters used in this study.

**Methods and Protocols**

Brief descriptions of the main methods used in this study are included here. Details of computational methods, algorithms, and models are provided in Appendix Supplementary Methods, following the same order of related sections in the main text.

### Metabolic network models
#### Generic *C. elegans* model, iCEL1314
The original reconstruction (Yilmaz & Walhout, 2016) was manually updated by the addition of new reactions, deletion or modification of existing reactions, or changes in reaction localization between cytosol and mitochondria. The basis of each modification is explained in Table EV1 with notes and titles of issues that are explained in Appendix Supplementary Methods. The targeted functionality of each modification was verified by FBA. The integrity of the updated model was checked with the help of MEMOTE (Lieven *et al*, 2020). The reactions and metabolites of iCEL1314 are presented in Tables EV2 and EV3, respectively.

### Dual-tissue model
Reactions of iCEL1314 were used to build a dual-tissue metabolic network model of four compartments: lumen (*L*), intestine (*I*), other tissue (*X*), and extracellular space (*E*) (Fig 2A). Internal reactions of the model (i.e., those taking place in cytosol or mitochondria only) were included in both *I* and *X* compartments, except for reactions that depend on bacterial degradation, which are represented only in *I*. Bacterial food and side nutrients (see below) and extracellular reactions for the intestine were placed in *L*. Transportable and exchangeable metabolites and extracellular reactions for the *X* tissue were placed in *E*. Transport reactions were distributed to exchange metabolites between *I* and *L* and between *E* and *X*. Exchange reactions were used in *L* to bring in nutrients from the environment, and in *E* to secrete by-products to the environment, with the exception of oxygen, water, orthophosphate, and protons, which could be both taken up and secreted through *E*.

### Experimental data
Tissue-level expression profiles were obtained from the reference study (Cao *et al*, 2017) as transcripts per million (TPM) and processed to divide genes, for each tissue, into four sets of expression levels: high, moderate, low, and rare.

### Categorization of genes by absolute expression levels

A histogram of average gene expression across tissues was generated using logarithm of TPM values at base 2 (Fig EV2). The result was a bimodal distribution that could be fitted by two superposed Gaussian curves (Appendix Equation S1), which represented a high expression subpopulation (HES) and a low expression subpopulation (LES). The means ($\mu$) and standard deviations ($\sigma$) that defined these curves were used to set thresholds for high expression ($> \mu_{HES}$), low expression ($< \mu_{LES} + \sigma_{LES}$), and rare expression ($< \mu_{LES}$). Then, for each tissue, genes were categorized accordingly (Appendix Fig S1).

### Gene categorization by relative expression levels

Moderately expressed genes were reevaluated with a heuristic algorithm that categorized some of these genes as highly or lowly expressed depending on their relative expression levels in different tissues. This algorithm evaluates one gene at a time. First, TPM values of the queried gene are listed for all seven tissues in an ascending order. Thus, expression level increases from one tissue to the next in six intervals. The fold change (*FC*) difference in each interval is separately evaluated. If *FC* is > 4 in the first interval or 1.5 in others, the gene is labeled as highly expressed in tissues after the interval evaluated. Likewise, if the expression level increases by 4 in the last interval or 1.5 in others, the gene is labeled as lowly expressed in tissues before the interval. If a tissue expression is inconsistently labeled as high and low during the evaluation of different intervals, then it was maintained as moderate. To minimize the effect of noise at low expression levels, the jump in TPM value in an interval was considered significant only if the greater value was higher than a threshold ($\tau_{rel} = \mu_{HES} - \sigma_{HES}$). Also, moderate expression levels were recategorized as high only if they were above this threshold, and as low only if they were below it. The general form of this algorithm is explained in detail in Appendix Supplementary Methods (Appendix Equations S2–S5) and is made available for use with any dataset (Appendix Table S3). Examples of genes recategorized as highly or lowly expressed in some tissues are provided in Fig EV3.

### Integration of model and experimental data

#### Constraint-based FBA and nutritional conditions

Data integration always included a steady-state mass balance of metabolites and constraint of flux between lower and upper bounds for each reaction of the model (Appendix Equations S5 and S6). Typically, reversible reactions were limited to values between −1,000 and 1,000 units of flux and irreversible reactions between 0 and 1,000, where the value 1,000 indicates a practically infinite flux and a negative value indicates a flux in the reverse direction of a reaction. A reaction that represents the usage of ATP for maintenance (RCC0005) (Yilmaz & Walhout, 2016) was constrained to carry at least 10 units of flux in both *I* and *X* compartments of the dual-tissue model (Table EV5).

The nutritional input of the model consisted of bacteria (a metabolite, BAC, which represents bacterial biomass), 243 side nutrients, and three storage molecules (Table EV3). When degraded in the intestine compartment, BAC provides a nutritional input consistent with the biomass composition of *Escherichia coli* (Yilmaz & Walhout, 2016). Side nutrients were selected from all exchangeable metabolites to provide the network with additional bacterial nutrients that may be quantitatively underrepresented in the

assumed bacterial biomass composition (e.g., individual amino acids) or nutrients that may be present only in the growth media (e.g., cholesterol). These additional nutrients and storage molecules were prevented from dominating the diet using a set of constraints and stoichiometric manipulations in uptake reactions (Appendix Equations S7–S12), which allow the limitation of the total uptake of side and storage nutrients to a small percentage of the bacterial intake by mass (see below for the choice of percentages). Together with an objective function that maximizes or minimizes a linear combination of fluxes (Appendix Equation S13), all applied constraints and mass balance equations constitute a regular FBA problem (Raman & Chandra, 2009) defined for the dual-tissue model. During data integration, bacterial intake was set unlimited, specific constraints and integer variables were added, and different objective functions were used, as will be explained below.

### Flux thresholds

To force reactions associated with highly expressed genes to carry flux during integration, the minimum value of a significant flux should be determined *a priori* (Shlomi *et al*, 2008). This threshold, designated $\varepsilon$, was set at 0.01 for > 90% of model reactions associated with genes. Setting the same threshold for some other reactions was not feasible since forcing that much flux in these reactions would require excessively large flux values (i.e., those greater than the default boundary of 1,000 units) in some parts of the network or a very large bacterial intake flux ($\gg$ 1 unit). This is because some metabolites have very small stoichiometric coefficients in reactions that appear during bacterial digestion or biomass assembly. To avoid this scaling problem, the maximum flux capacity of each reaction was first determined in each available direction (forward or reverse) using FVA (Mahadevan *et al*, 2002). FVA uses FBA by setting the objective function as maximization (to get the maximum forward flux) or minimization (to get the maximum reverse flux) of the flux of a single reaction. For this FVA problem, nutritional input was arbitrarily constrained by one unit of bacterial intake and the uptake of side and storage nutrients were each limited to 1% of bacterial intake by mass. Then, the maximum value obtained for the flux of a reaction was divided by 2. If this number was smaller than the default value of 0.01, then it was used as the flux threshold for the pertaining reaction in the pertaining direction, or else, the default value was used.

### iMAT++

iMAT++ was modified from the iMAT algorithm (Shlomi *et al*, 2008). iMAT++ follows the steps below to integrate a metabolic model with categorized experimental data:

- Define a binary integer variable ($y_i$) for each reaction that depends on rarely expressed genes, such that $y_i$ is 1 if the reaction has no flux and 0 otherwise (Appendix Equation S14). Flux through reactions identified this way is going to be blocked in as many cases as possible. These reactions make the reaction set $R_{OFF}$.

- Define binary variables also for each reaction associated with highly expressed genes but not dependent on lowly or rarely expressed genes ($y_i^f$ for the forward and $y_i^r$ for the reverse direction), such that $y_i$ is 1 if the reaction carries flux greater than $\varepsilon$ in the pertaining direction and 0 otherwise (Appendix Equations S15 and S16). These reactions will be promoted to carry flux and make the set $R_{ON}$.

- Define an integer variable ($z_i$) for each highly expressed gene that is associated with at least one reaction in $R_{ON}$. These genes are expected to be associated with flux-carrying reactions and make the set $G_{ON}$. Constrain $z_i$ to be less than or equal to the sum of $y$ values for all $R_{ON}$ reactions associated with the gene (Appendix Equation S20). At the same time, constrain $z_i$ to be less than or equal to 1 (Appendix Equation S21). Thus, $z_i$ takes the value of 1 if at least one reaction associated with the pertaining gene carries flux, else it is 0.
- Perform FBA to maximize the sum of all $z$ values for $G_{ON}$ and all $y$ values for $R_{OFF}$ combined (Appendix Equations S22 and S19). This sum is designated as $Z_{fit}$ and its maximum value as $Z_{fit,max}$. Constrain $Z_{fit}$ at $Z_{fit,max}$ for subsequent steps.
- Perform FBA to minimize total flux of reactions that depend on lowly or rarely expressed genes (Appendix Equations S23 and S24). The total flux of these reactions is designated as $Z_{low}$ and its minimum value as $Z_{low,min}$. Constrain $Z_{low}$ to be less than or equal to $Z_{low,min}$ plus a small tolerance value ($\delta_{low}$) for subsequent steps. The $\delta_{low}$ value used in this study was 1E-5.
- Perform FBA to minimize total flux of all reactions (Appendix Equations S25 and S26). The total flux is designated as $Z_{total}$ and its minimum value as $Z_{total,min}$. Constrain $Z_{total}$ to be less than or equal to $Z_{total,min}$ plus a small tolerance value ($\Delta_{total}$) for subsequent steps. The $\Delta_{total}$ value used in this study was $0.05 Z_{total,min}$. The flux distribution obtained in this step defines PFD (Fig 2B).
- Determine active metabolites in the previous flux distribution, i.e., those with a non-zero product sum of coefficients and absolute fluxes of associated reactions. Then, determine zero flux reactions that are associated with only highly expressed genes and that have only active metabolites as reactants (reversible reactions that do not carry flux are evaluated separately in either direction). These are latent reactions.
- To force latent reactions to carry flux, perform FBA to maximize the sum of the $y$ values (see above) of latent reactions determined (Appendix Equations S27 and S28). This sum is designated as $Z_{latent}$ and its maximum value as $Z_{latent,max}$. Constrain $Z_{latent}$ at $Z_{latent,max}$ for the next step.
- Perform FBA to minimize the total flux of all reactions as before (Appendix Equation S29). Since the resulting flux distribution may produce more latent reactions, iteratively go through the last three steps (including this) until no further latent reactions are added.
- The last flux minimization step produces the OFD (Fig 2B).

### Integration in the dual-tissue model context

Steps of integrating the dual-tissue model with experimental data are as follows:

- Constrain the uptake of stored metabolites (TAG, glycogen, and trehalose) as a percentage of bacterial intake by mass (see above for the method of constraining). An arbitrary upper limit of 1% was used for all stored metabolites combined.
- Constrain the uptake of side nutrients as a percentage of bacterial intake by mass. The overall uptake of these nutrients was limited by 2% of bacterial intake and the uptake of individual nutrients by 0.5% (see below for the choice of parameters).
- Integrate gene expression of the six non-intestinal tissues one by one with the model. Use all categorized genes in the target tissue to constrain reactions in $X$ and $E$ compartments. Use only rarely and lowly expressed genes in the intestine to constrain reactions

in the $I$ and $L$ compartments. Highly expressed genes in the intestine are ignored in this step.
- After flux distributions for all six non-intestinal tissues are obtained, calculate the sum of fluxes (from all six flux distributions) of reactions that carry material to and from the $X$ compartment and constrain the model with these fluxes for the next step. An efficient method is taking a weighted sum of the fluxes through the metabolites of the $E$ compartment and adding exchange reactions that exchange these metabolites, which are then constrained with the calculated metabolite fluxes (Appendix Equations S30–S32). The weights of this flux sum should reflect the differences in the metabolic activity of tissues (e.g., due to the differential tissue size). In this study, unique molecular identifiers (UMI) provided by the reference study (Cao et al, 2017) were used to approximate the relative activity of the tissues (Appendix Equation S33).
- Finally, integrate all categorized intestine genes with $I$ and $L$ compartments, while the flux to and from the $X$ compartment is taken into account by the constraints from the previous step.

Since all tissues are taken into account in the last step, the integration from that step shows the overall flux load and nutritional requirements. To warrant a bacterially dominated diet, different side nutrient uptake rates were tested for the second step. Because the intake of bacteria itself is not constrained, strict limitation of side nutrients as a percentage of bacterial intake results in the excessive intake of bacteria to allow sufficient uptake of side nutrients for the integration, as a result of which bacterial biomass is wasted and the ratio of side nutrients within the used resources is increased. Based on trials with different ratios (Appendix Fig S3A), the overall uptake of side nutrients was limited by 2% of bacterial intake and the uptake of individual metabolites by 0.5%. With this setting, bacterial nutrients dominated the resources used by about 95%.

### Determination of tissue metabolic networks

Whether a reaction was part of the metabolic network of a tissue was determined based on OFD (see above) and FVA (Mahadevan et al, 2002). Reversible reactions were evaluated separately in each direction. The algorithm is applied to each reaction as follows:

- Perform FVA: Use FBA, with constraints from iMAT++ integration for the tissue, to determine the maximum flux the reaction can take, which shows the flux capacity in the forward direction, and also (if applicable) the minimum flux the reaction can take, the negative of which shows the flux capacity in the reverse direction (Appendix Equation S35).
- Use the flux of the reaction from integration (i.e., in OFD for the tissue of interest) and the flux capacity of the reaction calculated by FVA to determine whether the reaction is in the tissue network:
  - If the reaction has non-zero flux in the tissue OFD, then it is part of the tissue metabolic network. These reactions were labeled as OFD (Tables EV7 and EV8).
  - If the reaction carries no flux in OFD, but has non-zero flux capacity based on FVA, then it carries flux in alternate solutions of integration for the tissue of interest and is therefore part of the tissue network. These reactions were labeled ALT.

○ If the reaction has zero flux capacity, then it does not carry flux in the solution space (SLNS) of the tissue of interest, and hence not part of the tissue network. These reactions were labeled NONE.

All FVA and OFD analyses were carried out with the dual-tissue model. However, tissue networks were eventually defined in a single-tissue format (Tables EV7 and EV8) using the reaction states in the $I$ and $L$ compartments for the intestine, and those in the $X$ and $E$ compartments for the other tissues. As an exception, if a reaction in the $I$ or $L$ compartment was labeled ALT for intestine integration but was not in SLNS for any other tissue integration, then it was considered as not part of the intestine network.

### Flux potential analysis
#### Algorithm
Flux potential analysis in a tissue is performed with the following steps:
- Divide reversible reactions into two separate reactions, one representing the forward direction and the other representing the reverse. The reverse reaction has the reactants and products swapped. Therefore, all reactions in the modified model have only a forward direction.
- Choose a target reaction. If the analysis is targeting a metabolite, choose a demand or export reaction that drains the metabolite to analyze its production potential, or a sink or import reaction that introduces the metabolite to analyze its consumption potential. If these reactions do not exist, insert a demand or sink reaction to the model accordingly.
- Calculate gene expression coefficients ($c$) for each reaction for the tissue of interest (see below). This calculation is done only once for all FPA performed for a particular tissue. For the dual-tissue model, expression coefficients in $I$ and $L$ compartments are always those based on intestine gene expression, while the coefficients of $X$ and $E$ reactions are variable.
- Calculate the metabolic distance ($d$) of each reaction in the network from the target reaction (see below).
- If the target reaction is part of a reversible reaction in the unmodified model (see above), block the reaction that represents the other direction in the original model. This step prevents loops.
- If the target reaction is a sink, demand, or transport (to or from extracellular space) reaction, block all other reactions of these types that act on the same metabolite. This step prevents short cuts during metabolite-level analyses.
- Block all reactions that are not part of the tissue network (see above for the determination of tissue networks).
- Perform FBA such that:
  ○ The objective function is the maximization of flux in the target reaction (Appendix Equation S36).
  ○ All reaction fluxes are constrained between 0 and 1000 (Appendix Equation S37).
  ○ A weighted sum of all fluxes is less than or equal to a constant allowance ($a$) (Appendix Equation S38). The weight of a reaction in this sum is a function of the gene expression coefficient for that reaction and its metabolic distance from the target (Fig 5A, Appendix Equation S39). The allowance used in this study was 1.
- The maximum flux obtained from the FBA step gives the flux potential (FP) of the target reaction in the tissue of interest.
- Calculate FP for the super system, i.e., for the same target reaction when gene expression coefficients are 1 for all reactions associated with genes.

- The ratio of FP calculated for the tissue to that calculated for the super system gives the relative flux potential (rFP) of the target reaction in the tissue of interest.

In the dual-tissue model context, reactions in $I$ and $X$ compartments were not targeted when FPA was performed on non-intestinal tissues and intestine, respectively. As a specific rule, during FPA for the intestine, all reactions in the $X$ compartment were blocked.

#### Calculation of gene expression coefficients
Tissue expression profiles (i.e., TPM values of a gene for the seven tissues) were used to determine gene expression coefficients of each reaction for each tissue. If a reaction was associated with only one gene, the profile was first normalized by the maximum value, and then the reciprocal of the normalized value for a tissue defined the coefficient of the reaction in that tissue (Appendix Equation S40). If multiple isozymes formed the reaction GPR, then the sum of TPMs for all isozymes was used to first generate a cumulative expression profile, and then normalization and derivation were carried out with this profile (Appendix Equations S40 and S41). If different proteins were involved in the GPR as in a protein complex, then the coefficient was derived for each protein, and the maximum value was used (Appendix Equations S40 and S42).

Reactions that are not associated with any genes were typically assigned an expression coefficient of 1. Exceptions included bacterial intake and degradation reactions and all exchange reactions ($E$ compartment), which were assigned a zero coefficient for all tissue evaluations. Also, transport reactions between $I$ and $E$ compartments were assigned a zero coefficient when the target reaction is in the $X$ compartment. This rule prevents the penalization of a non-intestinal tissue for having to transport a metabolite through a longer route than the intestine.

#### Metabolic distance
Metabolic distance between two reactions is defined as the length of the shortest path to reach from one reaction to the other. Calculation of guaranteed shortest paths is not possible for large metabolic networks for which exhaustive searches are not computationally feasible (Frainay & Jourdan, 2017). In this study, an algorithm was developed to effectively find the distance between reaction pairs by a non-exhaustive search (Appendix Table S3). The distance between all reaction pairs of the network can be calculated using this algorithm as follows:
- Convert the metabolic network into a network of only irreversible reactions (see the first step of the FPA algorithm above).
- Convert this metabolic network to a reaction tree such that each node represents a particular reaction and is connected to two types of other nodes called "to" nodes and "from" nodes. The "to" nodes represent reactions that consume the products of this reaction and the "from" nodes represent ones that produce its reactants. To prevent creating short cuts in the network, ignore hub metabolites such as atp, nadh, and h2o when linking reaction nodes. In this study, 21 frequent metabolites were ignored.
- Given a query reaction $i$, first eliminate the reverse of this reaction in the network (i.e., if the reaction represents one of the two directions of an originally reversible reaction) and traverse the tree from this reaction to all others that are reachable. When a reaction is reached for the first time, fix the path to that reaction before traversing the tree further to others. Because of this path fixing,

the path from reaction *i* to any reaction *j* will cover the least number of steps possible, thereby showing the distance from reaction *i* to reaction *j*.

- Check for loops in the path from reaction *i* to reaction *j*. A loop occurs when a reversible reaction of the original metabolic network is traversed twice, i.e., when both the forward and the reverse reactions originating from a reversible reaction are part of the path found. Such paths are not valid and cannot be used for calculating the shortest distance. For all such cases, eliminate the first reaction in the path that creates a loop and find the shortest path from reaction *i* to reaction *j* again.
- Since the previous step can still produce an invalid path with loops, repeat that step until a valid path is found from reaction *i* to reaction *j*. The length of this path defines the distance from reaction *i* to reaction *j*.
- Apply the previous three steps to find the distance to all reactions reachable from reaction *i*. Use the maximum distance observed in the network for reactions that are not reachable starting from reaction *i*.
- Repeat the previous four steps using every reaction in the network as the query. The result is a square distance matrix that gives distance from any reaction *i* to any reaction *j* in the metabolic network.
- Due to the structural properties of metabolic networks, the distance from *i* to *j* is not necessarily equal to the distance from *j* to *i*. Set the final distance between *i* and *j* as the minimum of these two values. This step converts the distance matrix to a symmetric one. During FPA, the row of the distance matrix that represents the target reaction can be used to define the distance between this reaction and all network reactions (columns).

The loop corrections in the fourth and fifth steps do not warrant shortest paths to be found. However, such corrections were necessary for < 1% of all distances calculated and more than a hundred such cases were manually verified to have yielded the shortest distances. Thus, the compromise made in this algorithm to make the search for the shortest path computationally feasible had a negligible effect on the accuracy of the calculated distances.

The exclusion of hub metabolites such as NAD and ATP in the second step means that some redox and energy reactions may appear very distant to a target reaction, and therefore, the weight of their flux may be very low during FPA. This does not mean redox and energy balance can be neglected, as redox, mass, and energy balances are always established by FBA during FPA. However, the influence of the relative expression of redox and energy genes may be diminished by the large distances, depending on the target reaction and the distance order used.

## Data availability

The computer code, metabolic models, and datasets produced in this study are available in the following databases:

- Code for MERGE: GitHub repository created for this project (https://github.com/WalhoutLab/MERGE), which includes scripts that reproduces the results here, examples of integration of iCEL1314 with whole-animal datasets, and a preliminary integration of a human model (Swainston *et al*, 2016) with transcriptomic data from various human tissues (Uhlen *et al*, 2015).

- Metabolic models: iCEL1314 is available at the WormFlux website (http://wormflux.umassmed.edu/) (Yilmaz & Walhout, 2016), in BioModels (Chelliah *et al*, 2015), and in Tables EV2 and EV3. The dual-tissue model is available in Table EV5. Both models are also available in the GitHub repository for MERGE.
- Predictions: Tables EV7 and EV8 of this study.

**Expanded View** for this article is available online.

### Acknowledgements
We thank members of the Walhout Lab, especially Olga Ponomarova, Cedric Diot, and Hefei Zhang for valuable discussions, and Amy Holdorf for critical evaluations of the metabolic network model. This work was supported by grants from the National Institutes of Health GM122502 to A.J.M.W. and DK115690 to A.J.M.W. and F.S.

### Author contributions
LSY and AJMW conceived the work and wrote the paper. LSY designed and implemented MERGE with help from XL. XL reproduced the results in MATLAB environment, created the GitHub repository, and performed applications with human model. SN helped with metabolic network model updates. BF, XL, and FS reconstructed the ascaroside biosynthesis pathway and helped with metabolic network model updates. All authors carefully reviewed and edited the paper.

### Conflict of interest
The authors declare that they have no conflict of interest.

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
