## [Review Process File · Molecular Systems Biology]

Modeling tissue-relevant *C. elegans* metabolism at network, pathway, reaction, and metabolite levels

Safak yilmaz, Xuhang Li, Shivani Nanda, Bennett Fox, Frank Schroeder, and Albertha Walhout
DOI: 10.15252/msb.20209649

Corresponding author(s): Albertha Walhout (marian.walhout@umassmed.edu), Albertha Walhout (marian.walhout@umassmed.edu), Safak yilmaz (lutfu.yilmaz@umassmed.edu)

Review Timeline:

Submission Date:	22nd Apr 20
Editorial Decision:	8th Jun 20
Revision Received:	29th Jul 20
Editorial Decision:	8th Sep 20
Revision Received:	10th Sep 20
Accepted:	11th Sep 20

Editor: Maria Polychronidou

Transaction Report:

Thank you again for submitting your work to Molecular Systems Biology. I apologise for the somewhat slow process, but the reviewers had asked us to extend their deadlines due to the covid-19 situation. We have now heard back from the four referees who agreed to evaluate your study. Overall, the reviewers think that the proposed approach seems potentially relevant for modelling metabolism at the level of whole animals. They raise however a series of concerns, which we would ask you to address in a revision.

I think that the reviewers' recommendations are rather clear and there is therefore no need to repeat all the points listed below. Please let me know in case you would like to discuss any of the issues raised, I would be happy to schedule a call. Regarding the point of reviewer #2 who requests concrete evidence of the applicability of MERGE in other organisms, we think that extensive analyses in this direction seem outside the scope of the study. However, some level of support for the broader applicability of MERGE would significantly enhance the impact of the work and we would encourage you to include it. Finally, reviewer #4, who is not a computational biologist but an expert on *C. elegans* metabolism, finds the study difficult to read. We would not agree with the suggestion to remove the methodological/computational details from the main text, as they are an integral part of the study and relevant for the audience of MSB. We would only ask you to make sure that the parts of the study likely to be relevant for purely experimental scientists are easily accessible to them.

On a more editorial level, we would ask you to address the following.

REFEREE REPORTS

Reviewer #1:

The manuscript by Yilmaz et al. reports a tissue-level metabolic network (suitable for constraint-based flux simulations) for *C. elegans*. The network/model is based on an updated version of a previous model and integration of gene expression data (to achieve tissue-level specificity). To build the model, the authors also improve upon previous algorithms and introduce a concept of Flux Potential Analysis (FPA) towards obtaining better discrimination between the (metabolic) functions of different tissues. The model assessment is done using known tissue metabolic phenotypes and a comparison to human tissues is also presented.

Overall, the study, in my opinion, has potential to advance the field towards modelling whole-animal metabolism and the use of gene expression data to infer metabolic functionalities in a fine-grained manner. There are several points where the manuscript needs substantial improvements before this potential is clear/accessible.

(The manuscript would be much easier to read if non-common/technical abbreviations would be less used, esp. in figures)

1. FPA is a very interesting approach. It, however, is a heuristic and should be clearly marked as such together with the assumptions that underlie it. Importantly, the method inherently assumes a direct concordance between mRNA expression levels and flux. Though such an assumption might be "OK" at the network level, when applying with a focus on a particular reaction, it may lead to overfitting.
2. Another unclear yet important aspect for FPA is how highly connected hub metabolites (NADH, ATP etc.) are handled therein. Considering these means that the most reactions will not be much far from any other reaction in the network. On the other hand, ignoring these would be mean neglecting the redox balance.
3. The claims of novelty regarding methodology are, in my opinion, should be removed. Most

changes over existing algorithms could be seen as incremental and, in any case, it is better to leave it the readers to assess this.

4. Single cell data suffers from stochasticity and evidence of absence cannot be wholly relied upon. How was this handled? It seems from the Methods that aggregates per tissue were used; in that case the mentions of single cell in the text can be misleading.

5. The mention of different 'scales' in the title is rather confusing ("network, pathway, reaction, and metabolite scales"). A genome-scale network implicitly has all these scales and I did not see any reason why these should be spelled out explicitly.

6. The presented models should be tested for community standard using MEMOTE (Lieven et al., Nat Biotechnol. 2020).

7. In the section "agreement between gene expression and optimal flux distributions", on Page 11, it is stated that "...tissue-level expression data suggests that the shunt pathway is active only in intestine and hypodermis". It is not apparent how this conclusion can be reached based solely on the particular tissue-expression data. Based solely on these expression data, a more accurate statement is that the pathway could also be active in two additional tissues (gonads and neurons), albeit to probably a lesser extent compared to intestine and hypodermis (Figure 2C). The authors use this statement to (partly) exemplify the superiority of the IMAT++ compared to IMAT. Are there any other literature-based data that the pathway is indeed only active in these tissues that could further support IMAT++ results?

Along these lines, there is a discrepancy for gene "acdh-1" between Figure 2C and Figure EV4, for the tissue "gonads". In Figure 2C is depicted as "lowly expressed", but there is no depiction at all in Figure EV4.

8. Figure 2D, in the middle panel: how this comparison between IMAT and IMAT++ for rarely expressed genes/reactions was carried out? Given the information in Appendix-sup, on page 12, it is stated that, for IMAT "the gene categorization in Figure 1E was modified by merging rarely and lowly expressed genes into one group, named lowly expressed, as the original algorithm does not include a rarely expressed category".

9. Figure 2D, last panel: in the respective section of the main text, it is stated that "the average flux in reactions dependent on lowly expressed genes was greatly depleted...". In the Figure, no flux is depicted for lowly expressed genes - why is this considered a more accurate representation of the metabolic physiology? Expression levels and fluxes can diverge due to translational control, metabolic control, PTMs, enzyme efficiency differences etc.

10. Page 15, "also supports the hypothesis that hypodermis acts as liver in *C. elegans*". So, harboring more reactions means acting as liver?

11. Page 19-20. Using one fixed distance order that best fit solely one metabolite to subsequently calculate relative flux potentials for the rest of the reactions could potentially deter the appearance of other interesting features. Did the authors try to estimate the same for varying distance orders (on the lower scale) to see how/if the results deviate from each other?

12. Section "Systematic Analysis of Tissue Function..." pages 19-24. In this section overall, it is confusing whether the definition of "tissue-specific" reactions/metabolites indeed refers to uniquely

associated components for each tissue, or components more enriched to a specific tissue relative to the others. It seems that the word is sometimes used interchangeably with "tissue-enriched" components and "potential". Thus, the phrasing should be carefully re-evaluated by authors (i.e. replaced with the term "tissue-enriched").

13. Section "Systematic Analysis of Tissue Function..." pages 19-24. In addition to the description of the examples derived from the mammalian systems that could potentially be applied to *C.elegans* (described as "coherent" predictions in Figure 7), could a more systematic comparison (% explained by) with the mammalian equivalents be also estimated?

Minor comments:

- Section "Processing of the Gene Expression". How many "moderately-expressed genes" were finally re-categorized as highly and lowly expressed in the different tissues? Carrying the whole analysis without re-categorization what is the final outcome as a comparison?
- Page 3, introductory line, one could potentially rephrase to give an overall impact of metabolism's role, not solely connected to biomass and energy generation
- The "rarely expressed" gene category lacks of a solid definition in Appendix sup.
- Main text, page 12, "consistent with the fact that the body size of *C. elegans* increases dramatically as it proceeds through the different stages of larval development". The respective reference is missing
- Pages 20-23, some references are appearing in a superscript format.
- The heading of "summary of findings with MERGE" more accurately represents future directions for exploration rather a summary of all findings.
- Figure 1E, reporting the categories as a percentage could be more useful for comparison - with the respective number of genes reported in parentheses

Reviewer #2:

In this study the authors presented MERGE (Metabolic models Reconciled with Gene Expression), a novel computational pipeline that they used to predict tissue-relevant metabolic function at the network-, pathway-, reaction-, and metabolite-level based on single-cell RNA-sequencing (RNA-seq) data from the nematode *Caenorhabditis elegans*. Their analysis recapitulated known tissue functions in *C. elegans*, captured metabolic properties that are shared with similar tissues in human, and also provided predictions for novel metabolic functions. The method is quite interesting, and the predicted tissue-level metabolic functions are in good agreement with biological knowledge. However, the whole pipeline seems to be very specific and tailored for this study, and I have major concern about its applicability in more general cases.

Major concerns

1. The execution of the whole pipeline included many parameters and the criteria for selection of the parameters looks very specific to the specific case in this study. For instance, the author selected in some cases reaction specific cutoff of flux to define active reaction without sensitivity analysis. A show case of applying this pipeline in another multi-tissue or multi-cell type scenario is needed. There are FBA based studies available not only for human liver, adipose muscle tissues but also for different cancer cell lines so the authors simply can apply their method and validate their

findings.

2. The novelty and significance of the pipeline is not justified. The iMAT++ is an adapted version of iMAT as the author claimed, but the method seems more similar to INIT as both of the methods divide the gene expression to four different levels. The novel concept that brought up by the author is the latent reactions, but the method should be compared with other similar methods developed in several case studies.

3. A systematic evaluation of the performance between the here presented and previously developed algorithm is needed. Please see paper below.

<https://journals.plos.org/ploscompbiol/article?id=10.1371/journal.pcbi.1003580>

4. In order to publication of such method in MSB, systemic evaluation is also needed to verify the agreement between gene expression and optimal flux distributions.

Reviewer #3:

Summary: MERGE is a computational pipeline that allows for the construction of a dual-tissue-model. The method seeks to elucidate metabolic exchange between the intestine and one other tissue, which in this manuscript is done in the context of *C. elegans*. MERGE integrates transcriptomic data to create predictions at the reaction, pathway, and tissue function level. It does so through the application of multiple novel algorithms including IMATT++, ATL, and FPA. Through these algorithms MERGE is able to account for alternative optimal flux pathways as well as allow flux through reactions that are supported by lowly expressed genes. The methods developed in this paper offer a novel approach to tissue-specific model building within a whole-body context, which will contribute significantly to the field. However, the paper requires a more thorough discussion of the rationale used to develop this computational pipeline.

Major points:

- Reasoning behind updates to the *C. elegans* Metabolic Network Reconstruction need more explanation. There are no quantifications as to how these changes have improved the model or made it more biologically relevant
- Statistical analysis for gene expression categorization remained ambiguous even in supplementary information provided
- The first part of the IMATT++ update workflow that works to solve rare/low expression gene flux needs more explanation of reasoning
- FPA workflow needs more explanation of reasoning, such as an explanation of metabolic distance and distance order

Minor Points:

- Single cell RNA-seq analysis is referred to simply as RNA-seq, which is not the same as single cell RNA analysis
- Some anthropomorphic terms are utilized throughout (i.e. "encouraged" or "discouraged")
- The first paragraph of the results section appears to fit better in an introduction section of the paper
- The bacterial "diet" of *C. elegans* should be more clearly outlined for replication and clarity purposes
- Patterns used for EV2 should be edited for visual clarity

Reviewer #4:

Summary

This paper reports development of MERGE (Metabolic models Reconciled with Gene Expression), a computational prediction tool that integrates metabolomic and single-cell RNA-seq expression data to predict tissue-dominant metabolic networks in *C. elegans*. This is an update of a previous tool, with the main new innovation being the tissue-specific prediction capability. It is exciting that the new predictions for metabolic networks and flux fit with tissue-specific functions that have been described in *C. elegans* or humans, and with a recent hypothesis that the worm hypodermis fulfills some functions of the mammalian liver. It will now be of great interest to test new predictions that are reported here.

General remarks

- Are you convinced of the key conclusions?

Yes.

- What is the nature of the advance (conceptual, technical, clinical)?

This paper technically gives us capacity for predicting metabolite networks in tissues by using single cell-RNA-seq.

- How significant is the advance compared to previous knowledge?

The tissue-specific predictions will make possible many new studies, in which genetics or condition-specific analyses could be combined with metabolomics, expression analyses, and fresh predictions.

- What audience will be interested in this study?

It shows a new way how to anticipate the flux by tissues in a whole animal, so it will interest people who work with biological phenomena involved with metabolite flux such as growth, development, metabolic disease and aging. It will therefore appeal not only to systems biologists, but also will directly contribute to understanding and future studies of metabolism.

Major points

-Specific criticisms related to key conclusions

- The paper is written largely for systems biologists, and as a biologist I found myself breathing a huge sigh of relief once I got to the bottom of page 19. This may be a function of journal format, but readability for biologists would be greatly enhanced if much of the algorithm nitty gritty could be walled off in an experimental section or otherwise delimited so that a mere mortal biologist like myself could quickly zero in on the very interesting predictions and power of the approach.

- In this paper, it is concluded that MERGE can predict metabolite levels and flux in each tissue by using single cell RNA-seq data. The authors should acknowledge some important limitations and future horizons. They look only at total mRNA levels, and with current single-cell data may not yet be able to deconvolute with respect to individual transcripts. Some metabolic reactions may be regulated at the mRNA splicing level (this is known for pyruvate kinase in mammals, for example). They also miss ALL levels of possible post-transcriptional control, something that I suspect is important.

-Specify experiments or analyses required to demonstrate the conclusions

- Page 21, line 15, they claimed "biomass production alone consumes a significant portion of assimilated nutrients...". Is the proportion truly known? What happens during a short starvation? It is not necessary to do this, but if this is a gray area it should be acknowledged.

Minor points

-None

Response:

- We thank the Editor and the reviewers for their thorough review, and helpful comments and suggestions. We respond to all comments point by point in blue. We edited the paper and added further analyses as necessary.

Reviewer #1:

The manuscript by Yilmaz et al. reports a tissue-level metabolic network (suitable for constraint-based flux simulations) for *C. elegans*. The network/model is based on an updated version of a previous model and integration of gene expression data (to achieve tissue-level specificity). To build the model, the authors also improve upon previous algorithms and introduce a concept of Flux Potential Analysis (FPA) towards obtaining better discrimination between the (metabolic) functions of different tissues. The model assessment is done using known tissue metabolic phenotypes and a comparison to human tissues is also presented.

Overall, the study, in my opinion, has potential to advance the field towards modelling whole-animal metabolism and the use of gene expression data to infer metabolic functionalities in a fine-grained manner. There are several points where the manuscript needs substantial improvements before this potential is clear/accessible.

(The manuscript would be much easier to read if non-common/technical abbreviations would be less used, esp. in figures)

We thank the reviewer for the constructive comments. For improved clarity, we spelled out some instances of uncommon abbreviations (rFP, MOI, FVA) in the main text and in Figure 1, and spelled out OFD, ALT, and FVA in Figure 3 legend. We provide our response to other comments below.

1. FPA is a very interesting approach. It, however, is a heuristic and should be clearly marked as such together with the assumptions that underlie it. Importantly, the method inherently

assumes a direct concordance between mRNA expression levels and flux. Though such an assumption might be "OK" at the network level, when applying with a focus on a particular reaction, it may lead to overfitting.

We agree with the reviewer and mention the heuristic nature of FPA in the revised manuscript (page 26, line 19). Related to this and Comment #11 below, we performed sensitivity analyses for the distance order parameter used in FPA (page 26, line 20 to page 27, line 3).

We also agree that applying mRNA assumption at the single reaction level can be misleading. As elaborated in the original and revised manuscript, this is why FPA integrates mRNA expression levels not only at the target reaction but also in surrounding reactions.

2. Another unclear yet important aspect for FPA is how highly connected hub metabolites (NADH, ATP etc.) are handled therein. Considering these means that the most reactions will not be much far from any other reaction in the network. On the other hand, ignoring these would be mean neglecting the redox balance.

We agree with the reviewer that this aspect should be clear. When calculating the distance of relevant reactions, hub metabolites were not included to avoid short-cuts in the network. This was indicated in the *Appendix*, and it is now clearly stated in the newly added *Structured Methods* section (page 44, lines 18-20).

The redox and energy balance are not neglected, because FBA is always done during FPA, and FBA does not permit an imbalance in energy or reducing power. This is also indicated in the *Structured Methods* section (page 46, lines 11-17).

3. The claims of novelty regarding methodology are, in my opinion, should be removed. Most changes over existing algorithms could be seen as incremental and, in any case, it is better to leave it the readers to assess this.

We thank the reviewer for this point. MERGE consists of three modules, two of which are improvements of existing algorithms (iMAT++ and FVA), and one of which is novel (FPA). We have carefully mentioned this in the revised manuscript to avoid "overselling".

4. Single cell data suffers from stochasticity and evidence of absence cannot be wholly relied upon. How was this handled? It seems from the Methods that aggregates per tissue were used; in that case the mentions of single cell in the text can be misleading.

The reviewer is correct. We rephrased this in the revised manuscript where appropriate.

5. The mention of different 'scales' in the title is rather confusing ("network, pathway, reaction, and metabolite scales"). A genome-scale network implicitly has all these scales and I did not see any reason why these should be spelled out explicitly.

The reviewer is correct that the genome-scale model covers the entire network. However, the predictions we generate are at the levels mentioned. We changed the title

of the manuscript to “*Modeling tissue-relevant C. elegans metabolism at network, pathway, reaction and metabolite levels*” and also addressed this point in the revised *Introduction* (page 4, lines 17-23).

6. The presented models should be tested for community standard using MEMOTE (Lieven et al., Nat Biotechnol. 2020).

We thank the reviewer for the update; this paper was published when ours was already submitted. However, we now took the opportunity to better annotate our model in an updated SBML format and tested it using MEMOTE standards. We present the results in *Appendix Supplementary Methods* (section “Updates in *C. elegans* Metabolic Network Reconstruction”, subsection “Quality check of the reconstruction”). The reader is also informed about the check in the *Methods* (page 31, line 4).

We reevaluated the model regarding the topics where the model received a low score from MEMOTE. None of these violated modeling principles and were mostly due to incompatibility of the algorithm with the model structure. There were also mistakes in MEMOTE results as partially explained in the appendix (e.g., many reactions that are claimed to be not in mass balance by MEMOTE are actually in mass balance according to COBRApy, where the tested model file was actually created). In any case, we hope our discussion of MEMOTE sets an example for checking forward compatibility of older models to the new standards (our model was originally reconstructed 4 years ago).

7. In the section “agreement between gene expression and optimal flux distributions”, on Page 11, it is stated that “...tissue-level expression data suggests that the shunt pathway is active only in intestine and hypodermis”. It is not apparent how this conclusion can be reached based solely on the particular tissue-expression data. Based solely on these expression data, a more accurate statement is that the pathway could also be active in two additional tissues (gonads and neurons), albeit to probably a lesser extent compared to intestine and hypodermis (Figure 2C). The authors use this statement to (partly) exemplify the superiority of the iMAT++ compared to iMAT. Are there any other literature-based data that the pathway is indeed only active in these tissues that could further support iMAT++ results?

Along these lines, there is a discrepancy for gene “*acdH-1*” between Figure 2C and Figure EV4, for the tissue “gonads”. In Figure 2C is depicted as “lowly expressed”, but there is no depiction at all in Figure EV4.

We thank the reviewer for the thorough evaluation. The expression levels for *acdH-1* are 493 in intestine, 18.1 in hypodermis, 2.1 in neurons and 0.6 in gonad. The gonad bar was not visible in Fig. EV4 due to the scale and because it is right around the rare-to-low transition line (this is a very low expression level). This is now explained in the figure legend although the figure was also improved by making the threshold line thinner.

This profile suggests that *acdH-1* expression is mainly in the intestine but there is some significant expression also in hypodermis. These are in line with our observations with experimental assays, which we reference in the modified text (page 12, line 1).

8. Figure 2D, in the middle panel: how this comparison between iMAT and iMAT++ for rarely expressed genes/reactions was carried out? Given the information in Appendix-sup, on page 12, it is stated that, for iMAT "the gene categorization in Figure 1E was modified by merging rarely and lowly expressed genes into one group, named lowly expressed, as the original algorithm does not include a rarely expressed category".

Each comparison between the two algorithms were based on the results for the same set of genes or reactions. The absence of flux in rarely expressed reactions for iMAT++ was compared to the absence for flux in exactly the same set of reactions for iMAT. This is clarified in Figure 2 legend in the revised manuscript and the method is described in Appendix Supplementary Methods (not modified).

9. Figure 2D, last panel: in the respective section of the main text, it is stated that "the average flux in reactions dependent on lowly expressed genes was greatly depleted...". In the Figure, no flux is depicted for lowly expressed genes - why is this considered a more accurate representation of the metabolic physiology? Expression levels and fluxes can diverge due to translational control, metabolic control, PTMs, enzyme efficiency differences etc.

We agree that mRNA levels are not a perfect proxy for fluxes in all cases and discuss this in the revised manuscript. However, we assume that reactions dependent on genes expressed under a certain threshold of mRNA level do not carry significant flux. The same assumption exists for iMAT, which is compared to iMAT++ in Figure 2D. The difference is that iMAT minimizes the number of lowly expressed reactions that carry flux, while iMAT++ minimizes total flux in these reactions. Thus iMAT++ allows some flux in these reactions because the small enzyme expression can sometimes be important, in agreement with the reviewer's concerns. We see in the figure that, with both algorithms, lowly expressed genes carry relatively low-levels of flux on the average as assumed, but more so in iMAT++. To clarify these points we modified the sections that describe iMAT++ in relation to iMAT (page 9, line 18 to page 10, line 2; page 12, lines 9-15).

10. Page 15, "also supports the hypothesis that hypodermis acts as liver in *C. elegans*". So, harboring more reactions means acting as liver?

We removed this statement in the revised manuscript.

11. Page 19-20. Using one fixed distance order that best fit solely one metabolite to subsequently calculate relative flux potentials for the rest of the reactions could potentially deter the appearance of other interesting features. Did the authors try to estiMATE the same for varying distance orders (on the lower scale) to see how/if the results deviate from each other?

We thank the reviewer for bringing this up. We previously tried different distance orders but did not add a systematic analysis to the manuscript. We combined this analysis with those related to other reviewer comments about the sensitivity of MERGE results to other parameters and modules (Reviewer 1, minor comment #1; Reviewer 2, comments #1 and #2). The results are presented in a new subsection in *Results* ("Robustness and Usability of MERGE", page 26, line 20 to page 27, line 3), which

directs the reader to a new figure (Figure EV5) and section ("Robustness and Usability of MERGE") in *Appendix Supplementary Methods*. In brief, we get the best predictions with the current distance order, which was determined based on the analysis of collagen production potential of tissues, and an overwhelming majority of our results is robust to small differences in this parameter. However, for future applications careful tuning of this parameter may be necessary.

12. Section "Systematic Analysis of Tissue Function..." pages 19-24. In this section overall, it is confusing whether the definition of "tissue-specific" reactions/metabolites indeed refers to uniquely associated components for each tissue, or components more enriched to a specific tissue relative to the others. It seems that the word is sometimes used interchangeably with "tissue-enriched" components and "potential". Thus, the phrasing should be carefully re-evaluated by authors (i.e. replaced with the term "tissue-enriched").

We thank the reviewer for this important point. What we meant was these reactions/metabolites are enriched in one or two tissues. We revised this section and converted "tissue-specific" to "tissue-enriched".

13. Section "Systematic Analysis of Tissue Function..." pages 19-24. In addition to the description of the examples derived from the mammalian systems that could potentially be applied to *C.elegans* (described as "coherent" predictions in Figure 7), could a more systematic comparison (% explained by) with the mammalian equivalents be also estimated?

Unfortunately, there is no database of all metabolic functions for mammalian tissues, so a systematic comparison is not feasible at this point. We do think this is in part why systematic studies of tissue metabolism, as in our study, are needed for *C. elegans*. This point of view is briefly mentioned in the first paragraph of the new discussion section (page 27, lines 6-12).

Minor comments:

- Section "Processing of the Gene Expression". How many "moderately-expressed genes" were finally re-categorized as highly and lowly expressed in the different tissues? Carrying the whole analysis without re-categorization what is the final outcome as a comparison?

We added the answer to the first question to the revised *Appendix Supplementary Methods*: "On average, the ratio of moderately expressed genes (in the original set) converted to highly and lowly expressed genes by this algorithm was 11 +/- 7% and 22 +/- 3%, respectively (average +/- standard deviation for the seven tissues)."

We addressed the second question in relation to other reviewer comments about the sensitivity of MERGE results to parameters and modules (Reviewer 1, major comment #11; Reviewer 2, comments #1 and #2). The results are presented in the section entitled "Robustness and Usability of MERGE" in *Appendix Supplementary Methods* and in Figure EV5. In brief, the major findings of this study were still recovered without the addition of relative expression analysis, but the predictions were overall affected significantly. This is because some reactions of relatively highly expressed genes in a

tissue stop carrying flux in the tissue OFD without the relative expression analysis, and therefore, they lose the high confidence prediction status.

- Page 3, introductory line, one could potentially rephrase to give an overall impact of metabolism's role, not solely connected to biomass and energy generation

We rephrased this sentence in the revised manuscript (page 3, lines 2-5).

- The "rarely expressed" gene category lacks of a solid definition in Appendix sup.

We described these genes both by text and formulation in the revised *Appendix Supplementary Methods* (section entitled "Processing of the Gene Expression Dataset").

- Main text, page 12, "consistent with the fact that the body size of *C. elegans* increases dramatically as it proceeds through the different stages of larval development". The respective reference is missing

We added a reference to the revised manuscript.

- Pages 20-23, some references are appearing in a superscript format.

We corrected this mistake.

- The heading of "summary of findings with MERGE" more accurately represents future directions for exploration rather a summary of all findings.

We thank the reviewer for this point. We added the last paragraph of the previous section to this section. The title and the section in question should now match after this change.

- Figure 1E, reporting the categories as a percentage could be more useful for comparison - with the respective number of genes reported in parentheses

We considered this suggestion but placing both labels (percentage and number) is hard without complicating the figure or compromising the visibility of characters. Since percentages are visually delivered by slice size in pie charts, we opted to keep the numbers alone as the label.

Reviewer #2:

In this study the authors presented MERGE (Metabolic models Reconciled with Gene Expression), a novel computational pipeline that they used to predict tissue-relevant metabolic function at the network-, pathway-, reaction-, and metabolite-level based on single-cell RNA-sequencing (RNA-seq) data from the nematode *Caenorhabditis elegans*. Their analysis recapitulated known tissue functions in *C. elegans*, captured metabolic properties that are shared with similar tissues in human, and also provided predictions for novel metabolic

functions. The method is quite interesting, and the predicted tissue-level metabolic functions are in good agreement with biological knowledge. However, the whole pipeline seems to be very specific and tailored for this study, and I have major concern about its applicability in more general cases.

We thank the reviewer for the positive comments. We addressed specific concerns below.

Major concerns

1. The execution of the whole pipeline included many parameters and the criteria for selection of the parameters looks very specific to the specific case in this study. For instance, the author selected in some cases reaction specific cutoff of flux to define active reaction without sensitivity analysis. A show case of applying this pipeline in another multi-tissue or multi-cell type scenario is needed. There are FBA based studies available not only for human liver, adipose muscle tissues but also for different cancer cell lines so the authors simply can apply their method and validate their findings.

We thank the reviewer for these important points. With regard to reaction specific flux cutoffs, we provide results from extensive sensitivity analyses where we redid the entire integration by MERGE after changing flux thresholding rules. These are presented in *Results* in the new subsection named Robustness and Usability of MERGE (please also see the response to Reviewer 1, major comment #11 and minor comment #1). Our findings were not significantly sensitive to changes in flux thresholding, as explained in the relevant section and in the new Figure EV5.

With regards to the application of MERGE to other systems, we previously included an application to human tissue data in our GitHub repository. The full-scale application of MERGE to another model and dataset with detailed analyses is beyond the scope of our manuscript. However, in response to this comment, we now extended the human tissue analysis and provide examples of easily verifiable iMAT++ tissue-specific reaction clusters and FPA results. This is discussed in the revised *Discussion* (page 29, lines 1-13).

2. The novelty and significance of the pipeline is not justified. The iMAT++ is an adapted version of iMAT as the author claimed, but the method seems more similar to INIT as both of the methods divide the gene expression to four different levels. The novel concept that brought up by the author is the latent reactions, but the method should be compared with other similar methods developed in several case studies.

We thank the reviewer for bringing up INIT. Actually, since INIT is not a method for flux prediction, it is not an alternative to iMAT++ or FPA. Simply put, there is no way of using INIT for getting at the predictions we presented in this study. INIT is a method for network building, and that is where it is still potentially relevant to MERGE.

The network building step of MERGE uses FVA around iMAT++ by default, which may be slow for complicated models, such as the human model we used for the demonstration

of the applicability of MERGE in other systems (see above). We therefore tried INIT as a faster alternative for the network building step. However, results using INIT were similar to what we obtained by skipping FVA step altogether in MERGE, *i.e.*, using the entire network for every tissue during FPA calculations. Since INIT did not present any advantages over this approach, we cannot propose INIT as an alternative network building module. Instead, we developed two faster versions of our current network building algorithm to improve the usability of MERGE. We verified the performance of these algorithms in the new "Robustness and Usability of MERGE" section in *Results* (page 26, lines 4-18).

For our response to the rest of this comment, please see the response to Comment #3.

3. A systematic evaluation of the performance between the here presented and previously developed algorithm is needed. Please see paper below.

<https://journals.plos.org/ploscompbiol/article?id=10.1371/journal.pcbi.1003580>

We referred to this paper in our original manuscript. The bottom line is, none of the methods in there can be used to capture all tissue functions implicated by transcription in one go. They all relate solely to the first step (iMAT++). The reason we selected iMAT as a starting point is now explained in the revised *Results* (page 8, line 22 to page 9 line 8).

4. In order to publication of such method in MSB, systemic evaluation is also needed to verify the agreement between gene expression and optimal flux distributions.

This comment was addressed in the original manuscript in two ways. First, we have a specific figure (Figure 2D) that deals with the systematic evaluation of the agreement between optimized flux distributions from iMAT++ and gene expression data. Second, despite relatively good agreement and some useful predictions with iMAT++ (Figure 2D and Figure 3), we still needed to use the finer grained FPA method that can use continuous gene expression levels to generate high-confidence predictions. We hope additions in our revised *Results* and *Discussion* sections further clarifies these points (*e.g.*, page 27, line 16 to page 28, line 15).

Reviewer #3:

Summary: MERGE is a computational pipeline that allows for the construction of a dual-tissue-model. The method seeks to elucidate metabolic exchange between the intestine and one other tissue, which in this manuscript is done in the context of *C. elegans*. MERGE integrates transcriptomic data to create predictions at the reaction, pathway, and tissue function level. It does so through the application of multiple novel algorithms including iMAT++, ATL, and FPA. Through these algorithms MERGE is able to account for alternative optimal flux pathways as well as allow flux through reactions that are supported by lowly expressed genes. The methods developed in this paper offer a novel approach to tissue-specific model building within a whole-body context, which will contribute significantly to the field. However, the paper requires a more thorough discussion of the rationale used to develop this computational pipeline.

We thank the reviewer for the positive comments. We address the comments in detail below to further improve the clarity of the paper.

Major points:

- Reasoning behind updates to the *C. elegans* Metabolic Network Reconstruction need more explanation. There are no quantifications as to how these changes have improved the model or made it more biologically relevant

These changes are updates on the existing model based on new knowledge (e.g., sphingolipid composition of *C. elegans*), new annotations (many updates in KEGG and WormBase databases), one pathway reconstruction (ascaroside biosynthesis), and some corrections by manual curation (e.g., proper use of a malate-aspartate shuttle). The basis and nature of each change is indicated in Table EV1, with further details explained in Appendix, which is now more explicitly mentioned in the revised manuscript (page 30, line 17 to page 31, line 5). All newly added reactions can carry flux, which shows the success of added functionality. For example, based on tissue datasets, we can now infer that ascaroside biosynthesis is taking place mostly in the intestine, thanks to the added pathway. Annotation updates and corrections are necessary for the proper maintenance of our model.

-Statistical analysis for gene expression categorization remained ambiguous even in supplementary information provided

We revised the supplementary section to address this comment but did only minor changes as we do not know which parts are unclear. However, we now provide the code for our categorization algorithm which can take user defined parameters (e.g., whether two or three gaussian curves should be superposed and fitted to histogram of expression data) to determine rarely, lowly, moderately, and highly expressed genes as the output. This is indicated in the Appendix Supplementary Methods (Table S3) and mentioned in the new Methods section (page 33, lines 7-9).

-The first part of the iMAT++ update workflow that works to solve rare/low expression gene flux needs more explanation of reasoning

Brief explanations for this are in *Results*, while long and detailed explanations are in the Appendix Supplementary Methods. We revised both sections, and clarified parts we thought may be unclear (Main text: page 9, line 18 to page 10, line 2; page 10, lines 7-11; page 12, lines 8-15. Appendix Supplementary Methods: section "Integration Algorithm", subsection "iMAT++").

-FPA workflow needs more explanation of reasoning, such as an explanation of metabolic distance and distance order

We revised the relevant sections and added our reasoning for the inclusion of distance to the beginning of the FPA explanations (page 17, lines 6-9). In addition, the added

Structured Methods section has a subsection ("4. Flux Potential Analysis") that provides further information on the method including distance and distance order.

Minor Points:

-Single cell RNA-seq analysis is referred to simply as RNA-seq, which is not the same as single cell RNA analysis

We removed such references.

-Some anthropomorphic terms are utilized throughout (i.e. "encouraged" or "discouraged")

We revisited these sections and we think these verbs are appropriate in the way they are used.

-The first paragraph of the results section appears to fit better in an introduction section of the paper

We moved the paragraph to the introduction section in the revised manuscript.

-The bacterial "diet" of *C. elegans* should be more clearly outlined for replication and clarity purposes

This was done in our previous publication that introduced the iCEL model (Yilmaz and Walhout, 2016). The composition of bacterial diet is also provided in our website that harbors our model (<http://wormflux.umassmed.edu/modify.php>). In addition, we now have a section about the nutritional input in the new *Methods* (page 31, lines 1-13).

-Patterns used for EV2 should be edited for visual clarity

We made minor changes in this Figure to make the metabolic gene depictions more clear. We also directed the reader to Figure S1 in the figure legend of Figure EV2 for the full visualization of the histograms of metabolic genes alone, for every tissue.

Reviewer #4:

Summary

This paper reports development of MERGE (Metabolic models Reconciled with Gene Expression), a computational prediction tool that integrates metabolomic and single-cell RNA-seq expression data to predict tissue-dominant metabolic networks in *C. elegans*. This is an update of a previous tool, with the main new innovation being the tissue-specific prediction capability. It is exciting that the new predictions for metabolic networks and flux fit with tissue-specific functions that have been described in *C. elegans* or humans, and with a recent hypothesis that the worm hypodermis fulfills some functions of the mammalian liver. It will now be of great interest to test new predictions that are reported here.

General remarks

- Are you convinced of the key conclusions?

Yes.

- What is the nature of the advance (conceptual, technical, clinical)?

This paper technically gives us capacity for predicting metabolite networks in tissues by using single cell-RNA-seq.

- How significant is the advance compared to previous knowledge?

The tissue-specific predictions will make possible many new studies, in which genetics or condition-specific analyses could be combined with metabolomics, expression analyses, and fresh predictions.

- What audience will be interested in this study?

It shows a new way how to anticipate the flux by tissues in a whole animal, so it will interest people who work with biological phenomena involved with metabolite flux such as growth, development, metabolic disease and aging. It will therefore appeal not only to systems biologists, but also will directly contribute to understanding and future studies of metabolism.

We thank the reviewer for the general comments and the interest in our work. Our response to their concerns is below.

Major points

-Specific criticisms related to key conclusions

- The paper is written largely for systems biologists, and as a biologist I found myself breathing a huge sigh of relief once I got to the bottom of page 19. This may be a function of journal format, but readability for biologists would be greatly enhanced if much of the algorithm nitty gritty could be walled off in an experimental section or otherwise delimited so that a mere mortal biologist like myself could quickly zero in on the very interesting predictions and power of the approach.

We totally understand the reviewer's concern. In fact, we have been acutely aware of how difficult it is to make the paper useful and readable for both modelers (Yilmaz) and *C. elegans* biologists (Walhout). Therefore, we had tried to keep the flow as smooth as possible in the original submission. Unfortunately, however, it is not feasible to remove algorithmic explanations from the main text as the other reviewer comments show. We hope that clarifications in the revised manuscript made it more readable. We also hope the new Methods section provides a more readable, step-by-step summary of the MERGE pipeline for *C. elegans*- and other researchers, with details explained in Appendix for interested readers.

- In this paper, it is concluded that MERGE can predict metabolite levels and flux in each tissue by using single cell RNA-seq data. The authors should acknowledge some important limitations and future horizons. They look only at total mRNA levels, and with current single-cell data may not yet be able to deconvolute with respect to individual transcripts. Some metabolic reactions may be regulated at the mRNA splicing level (this is known for pyruvate kinase in mammals, for example). They also miss ALL levels of possible post-transcriptional control, something that I suspect is important.

We discussed these important points in the revised *Discussion* (page 29, lines 14-20).

-Specify experiments or analyses required to demonstrate the conclusions

- Page 21, line 15, they claimed "biomass production alone consumes a significant portion of assimilated nutrients...". Is the proportion truly known? What happens during a short starvation? It is not necessary to do this, but if this is a gray area it should be acknowledged.

These predictions cannot be expected to be quantitatively accurate, but we know that a large proportion of nutrients are assimilated into biomass during larval growth. We added references for this statement in the revised manuscript (page 13, lines 8-9).

Minor points

-None

Thank you for sending us your revised manuscript. We have now heard back from the two reviewers who were asked to evaluate the revised study. As you will see below, both reviewers are satisfied with the modifications made and are supportive of publication.

Before we formally accept the manuscript we would also ask you to address the following editorial issues

REFEREE REPORTS

Reviewer #1:

The authors have responded to my comments in (mostly) satisfactory manner. After the first round of review, I still think the manuscript is 'distractedly' focussed on methodology (I am not satisfied, e.g., by the authors' response on the title - but it is authors' choice). The conceptual novelty of the computational method is not high, several critical assumptions underlie the choices (esp. concerning mRNA-flux relation, thresholds used to draw the conclusions on tissue specific activity etc.; thus making it difficult to assess generality at this point). The claim of 'novelty' is still made in the abstract, which I think is not justified (for the reasons outlined in my previous review) and in any case it is not a good practice (several reputed journals actively discourage the usage of 'novel'). In my view, the value of the manuscript (which I think is great!) is more as a resource/illustration for applying integrative flux modelling to model animals.

Reviewer #3:

The authors addressed well each of the previous critiques and the paper is much improved. The paper will be a nice contribution to the field.

Response:

- Please find below our response to the minor comments from the Editor and the reviewers. Our responses are in blue.
- We addressed Reviewer 1's comment about novelty based on the Editor's guidance.
- We also made minor corrections and modifications in the text according to the editorial comments from the Editor.

Editor:

- In line with the comment of reviewer #1, we would ask you to remove the claim of novelty of the computational pipeline from the abstract. We do not think that removing the word novel is going to detract from the significance of the study, and we agree that this way the methodological advance is presented in a more balanced way.

We removed the word "novel" from the abstract of the revised manuscript.

Reviewer #1:

The authors have responded to my comments in (mostly) satisfactory manner. After the first round of review, I still think the manuscript is 'distractedly' focussed on methodology (I am not satisfied, e.g., by the authors' response on the title - but it is authors' choice). The conceptual novelty of the computational method is not high, several critical assumptions underlie the choices (esp. concerning mRNA-flux relation, thresholds used to draw the conclusions on tissue specific activity etc.; thus making it difficult to assess generality at this point). The claim of 'novelty' is still made in the abstract, which I think is not justified (for the reasons outlined in my previous review) and in any case it is not a good practice (several reputed journals actively discourage the usage of 'novel'). In my view, the value of the manuscript (which I think is great!) is more as a resource/illustration for applying integrative flux modelling to model animals.

We thank the reviewer for the constructive comments. To address the specific comment, we removed the emphasis of novelty from the abstract in the revised manuscript (Please see the response to the Editor's first comment, above).

Reviewer #3:

The authors addressed well each of the previous critiques and the paper is much improved. The paper will be a nice contribution to the field.

We agree that the manuscript improved significantly during the revision and we thank the reviewer for the encouraging comments.

Thank you again for sending us your revised manuscript. We are now satisfied with the modifications made and I am pleased to inform you that your paper has been accepted for publication.

Corresponding Author Name: L. Safak Yilmaz and Albertha J.M. Walhout

Manuscript Number: MSB-20-9649RR